# Axon-dependent expression of YAP/TAZ mediates Schwann cell remyelination but not proliferation after nerve injury

Matthew Grove[1,2], Hyunkyoung Lee[1,2], Huaqing Zhao[3], Young-Jin Son[1,2]*

[1]Shriners Hospitals Pediatric Research Center and Center for Neural Repair and Rehabilitation, Temple University, Philadelphia, United States; [2]Department of Anatomy and Cell Biology, Temple University, Philadelphia, United States; [3]Department of Clinical Sciences, Lewis Katz School of Medicine, Temple University, Philadelphia, United States

**Abstract** Previously we showed that YAP/TAZ promote not only proliferation but also differentiation of immature Schwann cells (SCs), thereby forming and maintaining the myelin sheath around peripheral axons (Grove et al., 2017). Here we show that YAP/TAZ are required for mature SCs to restore peripheral myelination, but not to proliferate, after nerve injury. We find that YAP/TAZ dramatically disappear from SCs of adult mice concurrent with axon degeneration after nerve injury. They reappear in SCs only if axons regenerate. YAP/TAZ ablation does not impair SC proliferation or transdifferentiation into growth promoting repair SCs. SCs lacking YAP/TAZ, however, fail to upregulate myelin-associated genes and completely fail to remyelinate regenerated axons. We also show that both YAP and TAZ are redundantly required for optimal remyelination. These findings suggest that axons regulate transcriptional activity of YAP/TAZ in adult SCs and that YAP/TAZ are essential for functional regeneration of peripheral nerve.

*For correspondence:
yson@temple.edu

**Competing interests:** The authors declare that no competing interests exist.

## Introduction

YAP (Yes-associated protein) and TAZ (Transcriptional coactivator with PDZ-binding motif), are paralogous transcription coactivators, chiefly known as potent stimulators of cellular proliferation in diverse developing (*von Gise et al., 2012*; *Zhang et al., 2012*; *Xin et al., 2013*; *Cotton et al., 2017*) and neoplastic (*Yu et al., 2015*; *Zanconato et al., 2016*; *Moon et al., 2018*) tissues. Consistent with this role, we and others have recently shown that YAP/TAZ promote vigorous proliferation of immature Schwann cells (SC) in developing peripheral nerves (*Poitelon et al., 2016*; *Deng et al., 2017*; *Grove et al., 2017*), and that overexpression of YAP/TAZ promotes abnormally excessive proliferation of mature SCs in adult peripheral nerves (*Mindos et al., 2017*; *Wu et al., 2018*). Unexpectedly, several groups demonstrated that YAP or YAP/TAZ promote differentiation of developing SCs by upregulating myelin-associated genes, thereby mediating developmental myelination (*Fernando et al., 2016*; *Lopez-Anido et al., 2016*; *Poitelon et al., 2016*; *Deng et al., 2017*; *Grove et al., 2017*). Our group additionally showed that YAP/TAZ are selectively expressed in differentiated myelin-forming SCs, and that they are required for maintenance of the myelin sheath in adult nerves (*Grove et al., 2017*). In many systems, YAP/TAZ shift to the cytoplasm concomitant with differentiation of developing cells, and the nuclear exclusion of YAP/TAZ is believed to be required for homeostatic maintenance of mature cells or tissues (*Varelas, 2014*; *Wang et al., 2018*). We were therefore intrigued to find that YAP/TAZ are nuclear and transcriptionally active in mature SCs maintaining peripheral myelination.

Building on these findings in developing and intact adult nerves, we now report on the role of YAP/TAZ in the regenerating nerve, in which SCs both proliferate and differentiate, as in developing

peripheral nerve. Following traumatic nerve injury, SCs in axotomized nerve rapidly dedifferentiate and proliferate as they convert to regeneration promoting 'repair' SCs (*Jessen and Mirsky, 2016*; *Tricaud and Park, 2017*). When repair SCs regain axon contacts, they re-differentiate to myelin-forming SCs, thereby restoring motor and sensory functions (*Fex Svennigsen and Dahlin, 2013*; *Stassart et al., 2013*). Strikingly, we found that YAP/TAZ disappear from denervated SCs but reappear in SCs as axons regenerate. Consistent with these observations, we found that YAP/TAZ are dispensable for SC proliferation after injury but required for remyelination of regenerated axons. These findings extend the role of YAP/TAZ to functional regeneration of injured nerves and suggest that SCs are dependent on axons for their transcriptional activity of YAP/TAZ.

## Results

### YAP/TAZ expression in Schwann cells is axon-dependent

Transcriptional regulation of SC proliferation and differentiation by YAP/TAZ depends on their nuclear localization. Nuclear YAP/TAZ in SCs of adult mice promote myelin gene expression, essentially maintaining peripheral nerve myelination (*Grove et al., 2017*). As the first step to determine the roles of YAP/TAZ in nerve repair, we examined spatiotemporal expression patterns of YAP/TAZ in adult mice after sciatic nerve crush injury (*Figure 1*). The nerve crush model evokes active axon degeneration in the distal nerve stump, while permitting new axons from the proximal nerve stump to regenerate through the crushed site within 1–2 days post injury (dpi) (*Kang and Lichtman, 2013*; *Jang et al., 2016*; *Frendo et al., 2019*). New axons then keep regenerating within the basal lamina tubes filled with SCs and their processes, at the speed of 1–4 mm/day, although the debris of degenerating axons and myelin is not yet completely removed. We killed these mice 1, 3, 6, 9, 12, and 24 dpi and immunostained proximal and distal nerve stumps of ~5 mm in length with an antibody specific for both YAP and TAZ (*Figure 1A*; at 12 dpi). At one dpi when distal axons remained largely intact, nuclear expression of YAP/TAZ in associated SCs was unchanged (*Figure 1B*; 1D-dstl). Strikingly, at three dpi when axon degeneration was robust and SCs lost axon contacts, YAP/TAZ were almost undetectable in SC nuclei (*Figure 1B*; 3D-dstl). Thus, SCs lose nuclear expression of YAP/TAZ as associated axons degenerate.

YAP/TAZ reappeared in the nuclei of SCs at 6 dpi, and these SCs were associated with regenerating axons (*Figure 1B*; 6D-dstl). By 12 dpi, as axon regeneration and maturation progressed further, more SCs exhibited strong nuclear expression of YAP/TAZ, comparable to that of SCs in the proximal neve stumps (*Figure 1B*; 12D-dstl, see also *Figure 1A*). These observations suggest that SCs upregulate nuclear YAP/TAZ, when they regain axon contacts as axons regenerate.

Notably, we frequently observed thin regenerating axons associated with SCs at 3dpi, but YAP/TAZ were undetectable in these SCs (*Figure 1B*; zoomed area of 3D-dstl). In contrast, SCs exhibiting strong YAP/TAZ at 6- and 12 dpi were associated with thick axons, which appeared large enough to be myelinated (i.e., 1>μm; *Figure 1B*; 6D-dstl, zoomed area of 12D-dstl). These observations suggest that YAP/TAZ are selectively upregulated in SCs associated with regenerating axons that have become large enough to be myelinated. Consistent with this notion, YAP/TAZ are expressed in myelinating, but not in non-myelinating, SCs (*Grove et al., 2017*).

Western blotting also revealed marked reduction of YAP and TAZ levels at 3 dpi, followed by rapid upregulation of TAZ levels (*Figure 1C*, *Figure 1—source data 2*). Notably, YAP levels remained low in nerve lysates at 12 dpi (*Figure 1C*). As cells other than SCs can affect overall YAP levels (*Gaudet et al., 2011*; *Stierli et al., 2018*), we next examined expression of YAP in SCs of crushed nerves by immunohistochemistry (IHC). We first verified that an antibody specifically recognized YAP, but not TAZ (*Figure 1—figure supplement 1A*). YAP is upregulated in many SC nuclei at 6 dpi and continues to be observed at 24 dpi (*Figure 1—figure supplement 1B*), demonstrating that YAP is also upregulated in SCs concomitant with axon regeneration.

The dramatic down- and upregulation of YAP/TAZ concurrent with axon degeneration and regeneration, respectively, suggest that SCs are dependent on axons for YAP/TAZ nuclear expression. To test further whether axons regulate YAP/TAZ expression in mature SCs, we next investigated if denervated SCs are capable of upregulating YAP/TAZ in the absence of regenerating axons. We

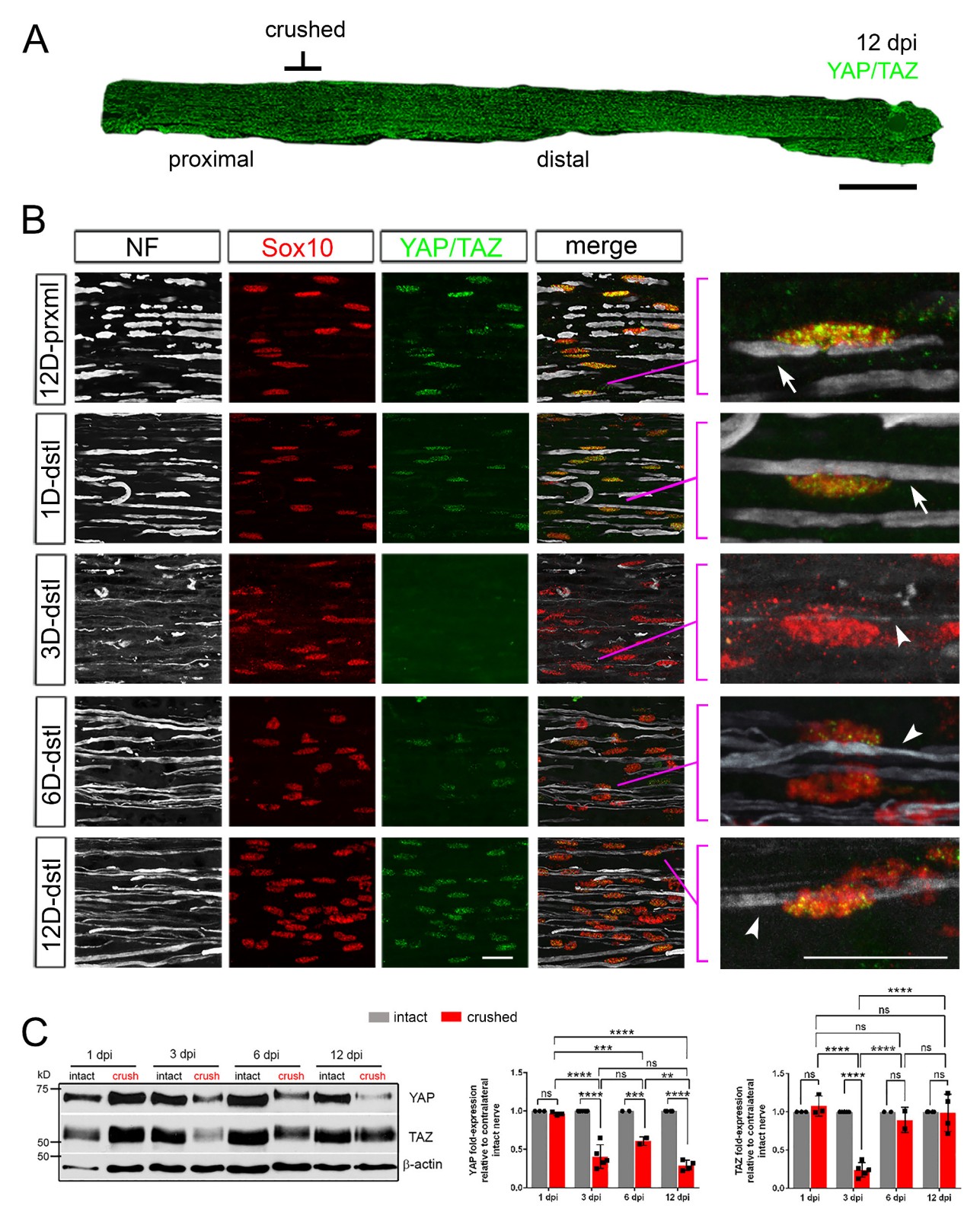

**Figure 1.** Loss and recovery of YAP/TAZ in Schwann cells after sciatic nerve crush. YAP/TAZ expression in crushed sciatic nerves of adult mice, shown by IHC (**A, B**) and Western blotting (**C**). Axons and Schwann cell (SC) nuclei are marked by neurofilament (NF) or Sox10, respectively. (**A**) A surgery schematic for nerve crush, which permits regeneration of axons into the distal nerve stump, illustrated by a low-magnification, longitudinal section of a sciatic nerve at 12 dpi, immunostained for YAP and TAZ. (**B**) Dramatic loss of YAP/TAZ in SC nuclei by three dpi, concomitant with axon degeneration,

*Figure 1 continued on next page*

*Figure 1 continued*

followed by upregulation of YAP/TAZ after six dpi, concomitant with axon regeneration. Right-most panels: zoomed area of merged images, as indicated, showing nuclear expression of YAP/TAZ in SCs associated with large diameter axons, before and after injury. Arrows point to large diameter axons in distal nerves before injury or 1 dpi, associated with YAP/TAZ+ SC nuclei. Arrowheads point to regenerating axons. Note that SC nuclei associated with a thin regenerating axon at 3 dpi do not express nuclear YAP/TAZ, but those in contact with a large diameter axon after 6 dpi do. Scale bars; 500 μm (**A**), 20 μm (**B**). (**C**) Western blotting of intact and crushed nerve lysates, showing loss of YAP and TAZ by 3 dpi, followed by full recovery of TAZ but not YAP by 12 dpi. Quantification of Western blots: n = 3–5 mice per experiment. ns = not significant, 2-way ANOVA. YAP: 1 dpi intact vs 1dpi crushed, p=0.9991; 1 dpi crushed vs 3 dpi crushed, ****p<0.0001; 1 dpi crushed vs 6 dpi crushed, ***p=0.0009; 1 dpi crushed vs 12 dpi crushed, ****p<0.0001; 3 dpi intact vs 3 dpi crushed, ****p<0.0001; 3 dpi crushed vs 6 dpi crushed, p=0.0652; 3 dpi crushed vs 12 dpi crushed, p=0.3479; 6 dpi intact vs 6 dpi crushed, ***p=0.0009; 6 dpi crushed vs 12 dpi crushed, **p=0.0018; 12 dpi intact vs 12 dpi crushed, ****p<0.0001. TAZ: 1 dpi intact vs 1 dpi crushed, p=0.9909; 1 dpi crushed vs 3 dpi crushed, ****P<0.0001; 1 dpi crushed vs 6 dpi crushed, p=0.6855; 1 dpi crushed vs 12 dpi crushed, p=0.9692; 3 dpi intact vs 3 dpi crushed, ****p<0.0001; 3 dpi crushed vs 6 dpi crushed, ****p<0.0001; 3 dpi crushed vs 12 dpi crushed, ****p<0.0001; 6 dpi intact vs 6 dpi crushed, p=0.9828; 6 dpi crushed vs 12 dpi crushed, p=0.9810; 12 dpi intact vs 12 dpi crushed, p>0.9999.

The online version of this article includes the following source data and figure supplement(s) for figure 1:

**Source data 1.** Source files for Yap and Taz Western graphs.
**Source data 2.** Time course of YAP and TAZ protein expression in WT nerves after sciatic nerve crush.
**Figure supplement 1.** Additional assessment of YAP expression in Schwann cells after nerve injury.

used a nerve transection injury model in which we completely cut one sciatic nerve and tied both ends of the transected nerve to prevent axon regeneration from proximal to distal nerve stumps. We examined these mice at 1, 3, 6, 9, 12, and 24 dpi. We first confirmed absence of regenerating axons in distal nerve stumps of these mice and found that YAP/TAZ become undetectable in SCs at 3 dpi (*Figure 2B*; 3D-dstl), concurrent with robust axon degeneration as observed after nerve crush injury. Notably, nuclear YAP/TAZ continued to be undetectable in SCs after 3 dpi (*Figure 2B*; e.g., 12D-dstl, see also *Figure 2A*), demonstrating that axons are required for YAP/TAZ upregulation in denervated SCs.

We also used an antibody specific for transcriptionally inactive, phosphorylated YAP (p-YAP), which is located preferentially in cytoplasm and exhibits perinuclear and membrane accumulation (*Grove et al., 2017*). We found that p-YAP became undetectable in SCs of transected/tied nerves by 12 dpi (*Figure 2C*). In contrast, p-YAP expression recovered in SCs of crushed nerves at 12 dpi (*Figure 2D*). These findings suggest that SCs are dependent on axons for both nuclear and cytoplasmic expression of YAP/TAZ.

## YAT/TAZ are dispensable for Schwann cell proliferation after nerve injury

SCs rapidly dedifferentiate and convert to repair SCs after nerve injury. During this transdifferentiation process, SCs begin to proliferate ~3 dpi (*Clemence et al., 1989*; *Jessen and Mirsky, 2016*; *Tricaud and Park, 2017*). Our observation that YAP/TAZ disappear in SCs by 3 days after axotomy raises the interesting possibility that YAP/TAZ are not involved in injury-elicited SC proliferation. Alternatively, levels of YAP/TAZ that are too low to be detected by IHC may be sufficient to promote transcription of the genes activating SC proliferation. To test these possibilities, we used an inducible knockout mouse (*Plp1-creERT2*; *Yap^{fl/fl}*; *Taz^{fl/fl}*, hereafter Yap/Taz iDKO) to inactivate YAP/TAZ selectively in SCs after nerve injury. We induced recombination at 6 weeks of age, completely transected and tied the sciatic nerve in one leg, killed the mice 5 days later when SCs actively proliferate, and compared SC proliferation in intact and transected nerves of WT and iDKO mice (*Figure 3A*, n = 3 mice per genotype). We first confirmed efficient ablation of YAP/TAZ in SCs by analyzing contralateral, intact nerves of iDKO mice (*Figure 3B and F*). We excluded mice with poor deletion (i.e., exhibiting YAP/TAZ in >20% SCs) from further analysis. Notably, pulse labeling with EdU indicated that the transected nerves of WT and iDKO contained similar numbers of dividing SCs in S phase (*Figure 3C and G*). Numbers of Ki67+ proliferating SCs (*Figure 3D and H*) and of total SCs (*Figure 3E and I*) were also similar in the transected nerves of WT and iDKO.

If adult SCs lacking YAP/TAZ in iDKO die or proliferate independently of axotomy, our analysis of injury-elicited SC proliferation might be confounded. To exclude this possibility, we examined contralateral, intact nerves of WT and iDKO mice at 12 dpi for SC proliferation and death (*Figure 3—*

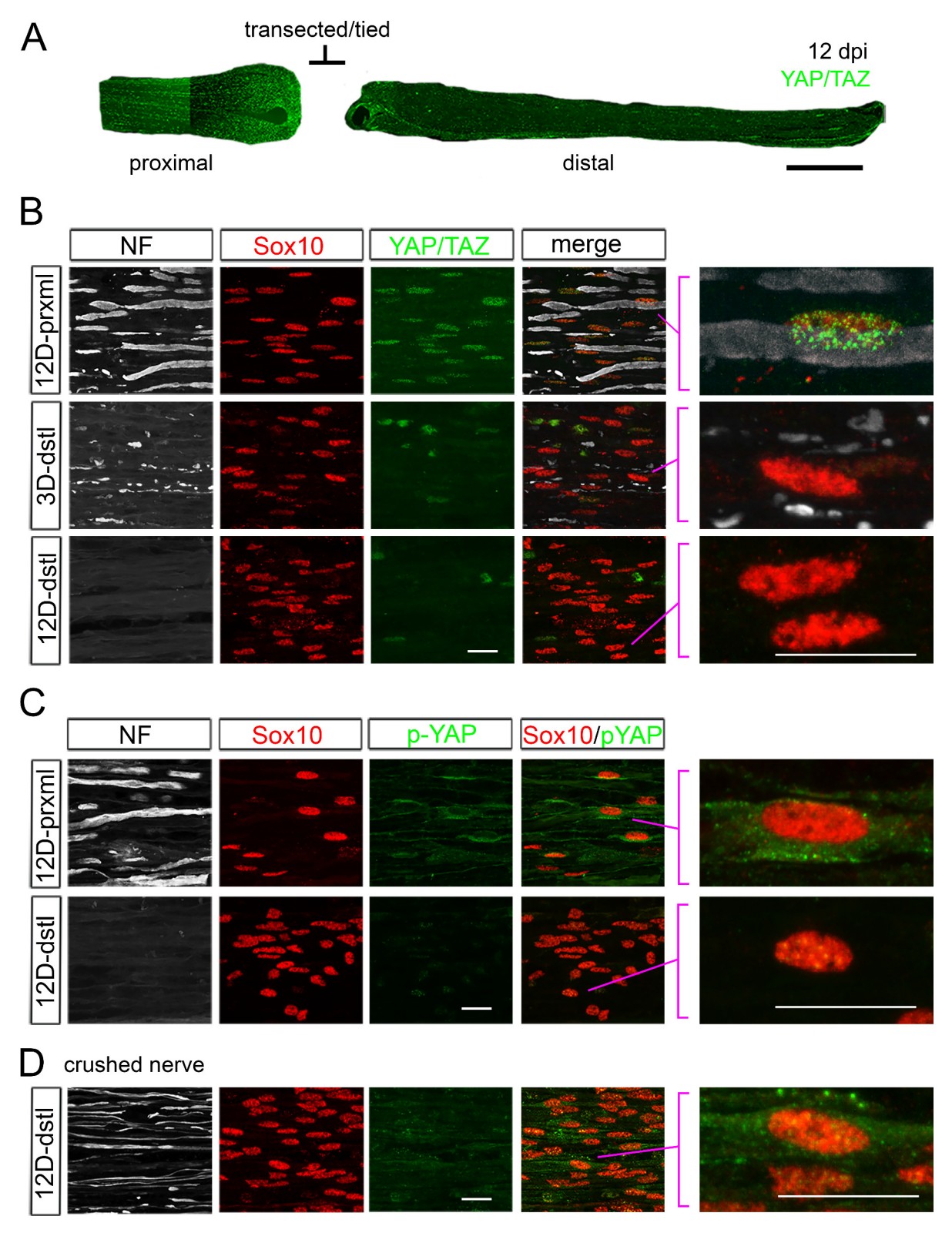

**Figure 2.** YAP/TAZ expression in Schwann cells after sciatic nerve transection. (**A, B, C**) YAP/TAZ expression in transected sciatic nerves of adult mice. Axons and Schwann cell (SC) nuclei are marked by neurofilament (NF) or Sox10, respectively. (**A**) A surgery schematic for nerve transection illustrated by a low-magnification, longitudinal section of a sciatic nerve at 12 dpi, immunostained for YAP and TAZ. Axon regeneration into the distal nerve stump was prevented by ligating the transected nerve stumps. (**B**) Complete loss of YAP/TAZ in SC nuclei at and after 3 dpi, concomitant with axon

*Figure 2 continued on next page*

*Figure 2 continued*

degeneration. Right-most panels: zoomed area of merged images, as indicated, showing that SCs do not upregulate YAP/TAZ in the absence of regenerating axons. (C) Cytoplasmic loss of phosphorylated YAP (p-YAP) in SCs of transected nerve. p-YAP was undetectable in axotomized SCs at 12 dpi. (D) Upregulation of p-YAP in SCs of crushed nerve. p-YAP was detectable in innervated SCs at 12 dpi. Right-most panel: zoomed area of merged image, showing a SC nucleus exhibiting perinuclear cytoplasmic p-YAP. Scale bars; 500 μm (A), 20 μm (B–D).

*figure supplement 1A*). Contralateral iDKO nerves contained neither EdU+ SCs (*Figure 3—figure supplement 1B*) nor apoptotic SCs, as assessed by TUNEL assays (*Figure 3—figure supplement 1C*). We also found that SC numbers did not differ significantly from those in intact nerves of WT mice (*Figure 3—figure supplement 1D and E*). Collectively, these results strongly indicate that YAP/TAZ do not regulate SC proliferation after nerve injury.

## SCs lacking YAP/TAZ convert to repair SCs and support axon regeneration

Next, we investigated if transdifferentiation to repair SCs proceeds normally in iDKO nerves after injury. We first examined expression of c-Jun, phosphorylated c-Jun (pc-Jun), p75 and Oct-6, which are associated with formation of repair SCs during nerve regeneration (*Scherer et al., 1994*; *Parkinson et al., 2008*; *Arthur-Farraj et al., 2012*; *Fontana et al., 2012*). Repair SC formation principally depends on the upregulation of c-Jun, which promotes expression of regeneration-associated genes (RAG), such as p75 neurotrophin receptor (NTR) (*Parkinson et al., 2008*; *Arthur-Farraj et al., 2012*; *Fontana et al., 2012*). Immunohistochemical analysis of transected sciatic nerves at five dpi showed that cJun, pc-Jun, p75 NTR and Oct-6 were all upregulated in denervated SCs of iDKO mice after nerve injury, as in WT mice (*Figure 4*, *Figure 4—figure supplement 1* and *Figure 4—source data 3*). Notably, injured nerves of WT and iDKO mice contained similar numbers of SCs expressing c-Jun (*Figure 4A and E*) and active pc-Jun (*Figure 4B and F*). There was minimal expression of c-Jun in contralateral, intact nerves of iDKO at five dpi (*Figure 4—figure supplement 1*). p75 NTR expression was also strongly upregulated in iDKO SCs, as in WT SCs (*Figure 4C and G*), and Oct-6 expression in WT and iDKO SCs did not differ (*Figure 4D and H*). Western blotting analysis confirmed upregulation of these proteins in injured nerves and validated the specificity of the antibodies used in the immunohistochemical analysis (*Figure 4—figure supplement 1* and *Figure 4—source data 3*).

SCs are essential for successful nerve regeneration (*Scheib and Höke, 2013*; *Jessen and Mirsky, 2016*). As the definitive test of whether iDKO SCs convert normally to repair SCs, we next examined if the absence of YAP/TAZ in SCs impairs nerve regeneration. Because *Yap/Taz* iDKO mice die ~14 days after tamoxifen treatment (*Grove et al., 2017*), we crushed sciatic nerves and analyzed them on 12–13 dpi. To minimize variability, we crushed nerves at the same site close to the sciatic notch and analyzed nerve segments immunohistochemically or ultrastructurally at the same distance distal to the injury (*Figure 5A*). An anti-β3 tubulin antibody, which identifies all axons, intensely labeled many axons that had regenerated through the ~10 mm long distal nerve stumps of iDKO mice (*Figure 5B and D*). These axons were as thick and numerous in iDKO as in WT nerves (*Figure 5B and F*, see also *Figure 7—figure supplement 1*). Similar numbers of axons were also present in contralateral intact nerves of WT and iDKO (*Figure 7—figure supplement 1*), indicating that there was no axon degeneration in intact nerves of iDKO at 12–13 dpi.

To confirm these findings, we examined transverse nerve segments 5 mm distal to the injury by TEM (Transmission Electron Microscopy). In this ultrastructural analysis, we took advantage of the fact that regenerating axons extend through the basal lamina (BL) tubes that surround SCs and their processes (*Scheib and Höke, 2013*; *Jessen and Mirsky, 2016*). We found that the percentage of BL tubes containing axons (single or multiple) was similar in WT and iDKO nerves (*Figure 5E and G*). Furthermore, the percentage of BL tubes containing axons large enough to be myelinated (i.e.,>1 μm) did not differ (*Figure 5H*). However, the large axons in iDKO nerves were more frequently accompanied by one or multiple, often thin, axons, which presumably represent transient collateral sprouts (*Figure 5E–d and I*).

Next, we examined iDKO nerves at an earlier time point after injury to investigate if axon regeneration might be delayed in the absence of YAP/TAZ in SCs. We analyzed longitudinal sections of crushed sciatic nerves from WT and iDKO mice at three dpi (*Figure 6*; n = 3 mice per genotype).

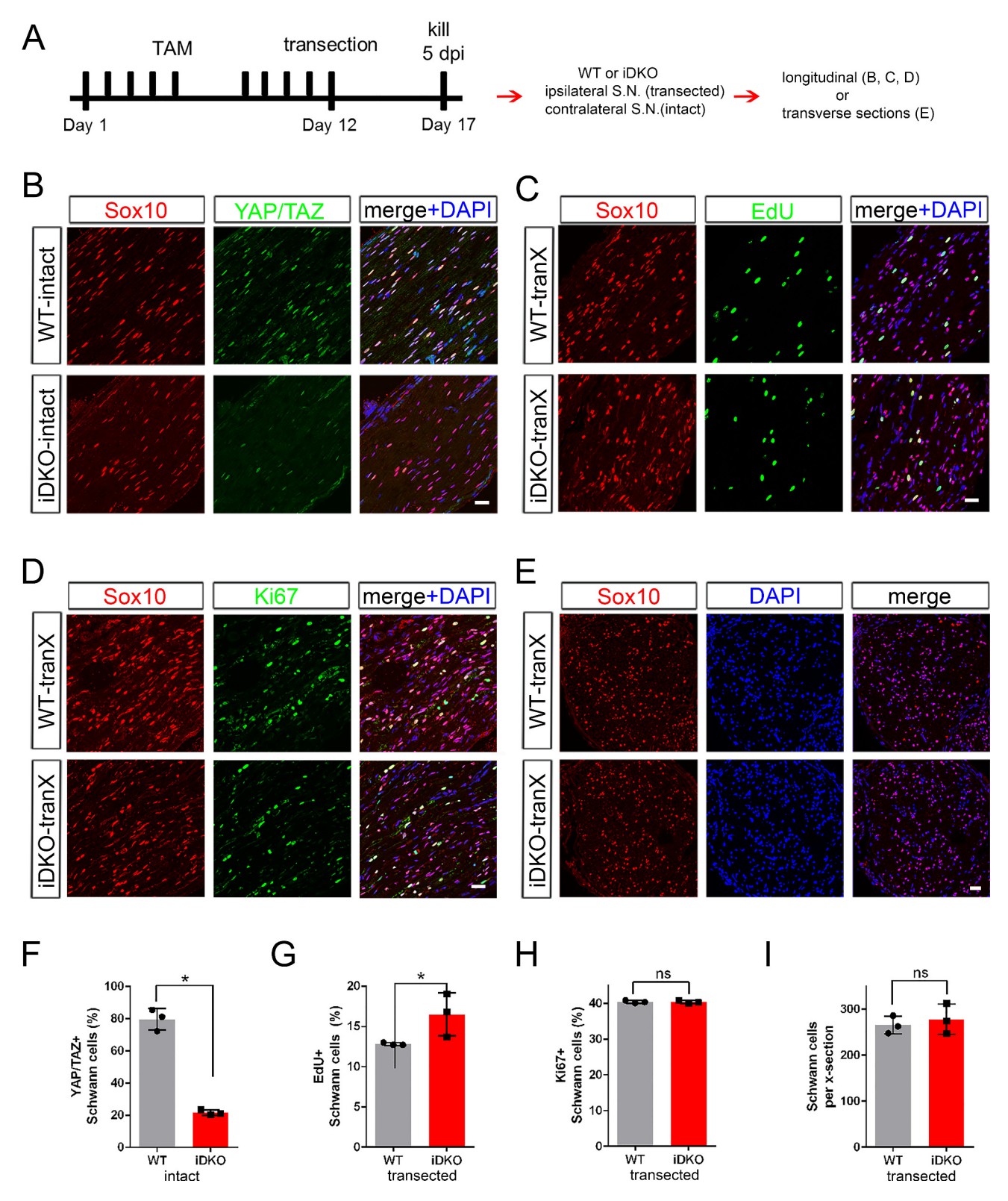

**Figure 3.** YAP/TAZ are dispensable for Schwann cell proliferation after axotomy. (**A**) Schematic showing timeline of tamoxifen injection, sciatic nerve transection and sacrifice of adult WT or Yap/Taz iDKO. (**B**) Longitudinal sections of intact sciatic nerves showing efficient deletion of YAP/TAZ in iDKO. SC nuclei are marked by Sox10 (red). All cell nuclei are marked by DAPI (blue). (**C**) Longitudinal sections of transected nerves of WT or iDKO showing SCs in S-phase of the cell cycle marked by EdU (green). (**D**) Longitudinal sections of transected nerves of WT or iDKO showing proliferating SCs marked

*Figure 3 continued on next page*

*Figure 3 continued*

by Ki67 (green). (**E**) Transverse sections of transected nerves of WT or iDKO showing SCs marked by Sox10 (red). (**F**) Quantification of SCs expressing nuclear YAP/TAZ in intact sciatic nerves of WT or iDKO. n = 3 mice per genotype, *p=0.0495, Mann-Whitney. (**G**) Quantification of EdU+ SCs in transected nerves of WT or iDKO. n = 3 mice per genotype, *p=0.0463, Mann-Whitney. (**H**) Quantification of Ki67+ proliferating SCs in transected nerves of WT or iDKO. n = 3 mice per genotype, ns, not significant, p=0.5127, Mann-Whitney. (**I**) Quantification of Sox10+ SCs in transected nerves of WT or iDKO. n = 3 mice per genotype. ns, not significant, p=0.8273, Mann-Whitney. Scale bars = 30 μm (**B–E**).

The online version of this article includes the following source data and figure supplement(s) for figure 3:

**Source data 1.** Source files for EdU$^+$ SC data.
**Source data 2.** Source files for Ki67$^+$ SC data.
**Source data 3.** Source files for graphs quantifying Yap/Taz+ SCs, EdU+ SCs, Ki67+ SCs, and total SCs.
**Figure supplement 1.** No Schwann cell proliferation or death in intact nerves of *Yap/Taz* iDKO at 12 dpi.
**Figure supplement 1—source data 1.** Source files for graph quantifying total SCs.

Because abundant debris of degenerating axons often confounded immunohistochemical identification of regenerating axons at this early time point, we selectively labeled regenerating axons with an antibody for superior cervical ganglion 10 (SCG10), which is rapidly and preferentially upregulated in sensory axons after injury (*Shin et al., 2014*; *Mogha et al., 2016*). Numerous axons reached ~4 mm distal to the crush site in iDKO, as in WT mice (*Figure 6A,B*). There was no significant difference in axon density measured 2 mm distal to the crush (*Figure 6C–E*), nor in the length of the longest axons (*Figure 6F*). Taken together, these results show that SCs lacking YAP/TAZ convert normally to repair SCs and support axon regeneration after injury.

## YAP/TAZ are required for Schwann cells to remyelinate axons

We have previously reported that developing SCs lacking YAP/TAZ arrest as promyelinating SCs, and are therefore unable to initiate myelin formation (*Grove et al., 2017*). To determine if adult SCs lacking YAP/TAZ can myelinate regenerating axons, we next analyzed the extent of myelination in the same iDKO nerves analyzed for axon regeneration on 12–13 dpi (*Figure 7A* shows the same nerves as *Figure 5B*). As expected, there was strong expression of myelin basic protein (MBP), a major structural component of the myelin sheath, in the crushed nerves of WT mice (*Figure 7A and B*). MBP immunoreactivity was also abundant in the contralateral, intact nerves of iDKO mice (*Figure 7A*; bottom panel), in which our previous ultrastructural analysis found segmental demyelination (*Grove et al., 2017*). In contrast, iDKO crushed nerves revealed remarkably little, if any, MBP immunoreactivity (*Figure 7A and C*, see also *Figure 7—figure supplement 1* for higher magnification images). Consistent with this immunohistochemical analysis, semithin (*Figure 7D*) and ultrathin sections processed for EM (*Figures 5E* and *7E*) contained many myelinated axons in WT but almost none in iDKO crushed nerves (*Figure 7F and G*). Moreover, iDKO SCs frequently surrounded and established a 1:1 relationship with large axons, but none of these axons exhibited a myelin sheath (*Figures 5E* and *7E*). These findings suggest that adult SCs lacking YAP/TAZ fail to remyelinate axons because they arrest at the promyelinating stage after injury.

## YAP and TAZ are functionally redundant and required for optimal remyelination

*Mindos et al., 2017* recently reported that expression of YAP, assessed by Western blotting, selectively increases after nerve injury in mutant nerves lacking Merlin in SCs, but not in WT nerves, whereas TAZ increases in both WT and mutant nerves. They also reported that elevated YAP levels prevent axon regeneration and remyelination, and that inactivation of YAP alone is sufficient to restore full functional recovery of the Merlin mutants (*Mindos et al., 2017*). These observations suggest that the function of TAZ in adult SCs may differ from that of YAP. We next examined axon regeneration and remyelination when SCs express YAP but not TAZ after injury. We reasoned that, if YAP prevents regeneration, regardless of expression levels (see Discussion), and if it differs functionally from TAZ, then we would find axon regeneration and remyelination to be poor.

Using a TAZ-selective tamoxifen inducible line to inactivate TAZ in SCs (*Plp1-creERT2; Yap$^{+/+}$; Taz$^{fl/fl}$*, hereafter *Taz* iKO), we crushed sciatic nerves unilaterally and compared the mutants to WT and *Yap/Taz* iDKO mice at 12 dpi. We first confirmed efficient ablation of TAZ and no compensatory

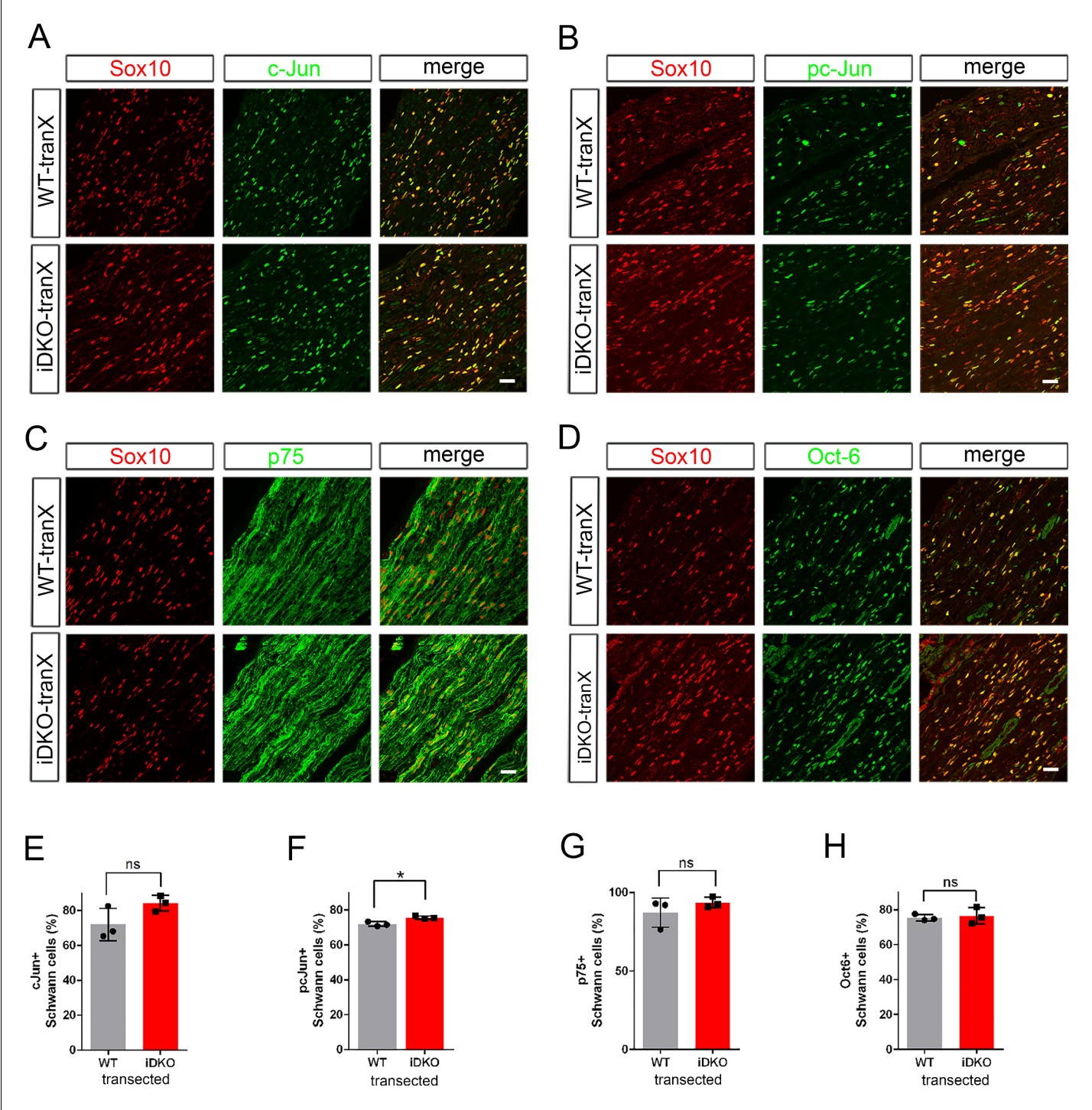

**Figure 4.** Schwann cells lacking YAP/TAZ transdifferentiate into repair Schwann cells. Longitudinal sections of transected sciatic nerves of WT and *Yap/Taz* iDKO immunostained by various markers of growth-promoting repair SCs at five dpi. SCs are marked by Sox10 (red). (**A**) Representative sections showing upregulation of c-Jun in iDKO SCs, as in WT SCs. (**B**) Upregulation of active phospho-S63 c-Jun in iDKO SCs, as in WT. (**C**) Upregulation of p75 in iDKO SCs, as in WT SCs. (**D**) Upregulation of Oct-6 in iDKO SCs, as in WT SCs. (**E**) Quantification of c-Jun+ SCs in WT and iDKO. n = 3 mice per genotype. ns, not significant, p=0.1266, Mann-Whitney. (**F**) Quantification of pc-Jun+ SCs in WT and iDKO. n = 3 mice per genotype. *p=0.0495, Mann-Whitney. (**G**) Quantification of p75+ SCs in WT and iDKO. n = 3 mice per genotype. ns, not significant, p=0.5127, Mann-Whitney. (**H**) Quantification of Oct-6+ SCs in WT and iDKO. n = 3 mice per genotype. ns, not significant, p=0.8273, Mann-Whitney. Scale bars = 30 μm (**A–D**).

The online version of this article includes the following source data and figure supplement(s) for figure 4:

**Source data 1.** Source files for c-Jun+ SC data.

*Figure 4 continued on next page*

*Figure 4 continued*

**Source data 2.** Source files for graphs quantifying c-Jun+ SCs, pc-Jun+ SCs, p75+ SCs, and Oct6+ SCs.
**Source data 3.** Western blotting analysis of repair Schwann cell markers.
**Figure supplement 1.** Western blotting analysis of repair Schwann cell markers.
**Figure supplement 1—source data 1.** Source files for graphs quantifying c-Jun, pc-Jun, p75 and Oct6 Westerns.

elevation of YAP levels in *Taz* iKO (*Figure 8A*, *Figure 8—source data 4*). We then used anti-β3 tubulin antibody to assess axon regeneration up to 15 mm distal to the crushed site (*Figure 8—figure supplement 1A,B*). We found that regenerating axons were as thick and numerous in *Taz* iKO, as in WT mice (*Figure 8—figure supplement 1C,D*). Axon density measured at 8 ~ 10 mm distal to the crush site showed no significant difference among WT, *Taz* iKO and *Yap/Taz* iDKO nerves (*Figure 8B*, *Figure 8—figure supplement 1E*).

Ultrastructural analysis of nerve segments at 5 mm distal to the injury revealed many BL tubes containing single or multiple axons in *Taz* iKO, as in WT (*Figure 8C*). These axon-containing BL tubes were as numerous in iKO as in WT and iDKO (*Figure 8D*). Counts of BL tubes containing axons large enough to be myelinated also did not differ (*Figure 8E*). Taken together, these results show that axons regenerated as robustly in *Taz* iKO as in WT and iDKO nerves, indicating that SCs expressing only YAP supported axon regeneration.

We also found that, whereas iDKO nerves contained no myelinated axons (e.g., *Figure 7D*), myelinated axons were frequent in *Taz* iKO nerves (*Figure 8C and F*), and G-ratios did not differ in *Taz* iKO and WT (*Figure 8G*), demonstrating that SCs expressing only YAP were capable of myelinating regenerated axons. Notably, however, a significantly smaller percentage of single axons were myelinated in *Taz* iKO than in WT (*Figure 8F*), indicating that remyelination is less advanced in *Taz* iKO nerves whose SCs express only YAP. Taken together, these results show that YAP, at normal levels (see Discussion), does not prevent axon regeneration or remyelination after injury, and that both YAP and TAZ are required for optimal remyelination.

## Redifferentiation of Schwann cells lacking YAP/TAZ

Following axon regeneration, denervated SCs that have regained axon contacts downregulate dedifferentiation-associated genes while upregulating genes promoting their differentiation (*Stassart et al., 2013*; *Quintes et al., 2016*; *Wu et al., 2016*). It is possible that YAP/TAZ-deficient iDKO SCs fail to myelinate regenerated axons because their capacity to carry out one or both processes is defective. To test if iDKO SCs correctly downregulate dedifferentiation-associated genes, we compared expression of c-Jun, Ki67 and Oct-6 by WT and iDKO SCs at 5 and 12 dpi after crush. The number of c-Jun+ SCs was markedly, but similarly, reduced in nerves of both WT and iDKO at 12 dpi (*Figure 9A,E*), and proliferating SCs were rare (*Figure 9B,F*). Oct-6 expression was also reduced in both WT and iDKO (*Figure 9C,G*), although it remained statistically higher in iDKO SCs. These results suggest that iDKO SCs are capable of downregulating dedifferentiation genes and withdraw gradually from dedifferentiation as like WT SCs.

Lastly, we examined expression of Krox 20 (also known as Egr2), the master transcription factor that drives myelin gene expression (*Topilko et al., 1994*; *Decker et al., 2006*). Notably, whereas WT SCs upregulated Krox 20 expression at 12 dpi, concomitant with remyelination, few if any iDKO SCs exhibited Krox 20 immunoreactivity (*Figure 9D,H*). These results suggest that iDKO SCs fail to myelinate regenerated axons at least in part due to failure to upregulate Krox 20.

## Discussion

Recent studies of SC-specific gene targeting consistently show that SCs lacking both YAP and TAZ are unable to proliferate properly and fail to myelinate developing peripheral nerves (*Poitelon et al., 2016*; *Deng et al., 2017*; *Grove et al., 2017*). It remains controversial, however, how YAP/TAZ loss results in complete amyelination of developing nerves and whether YAP/TAZ also play a role in myelin maintenance of adult nerves. Indeed, Poitelon et al., attributed developmental amyelination to the inability of immature SCs lacking YAP/TAZ to wrap around developing axons, a process called radial sorting (*Feltri et al., 2016*; *Poitelon et al., 2016*). In contrast, Grove et al. and

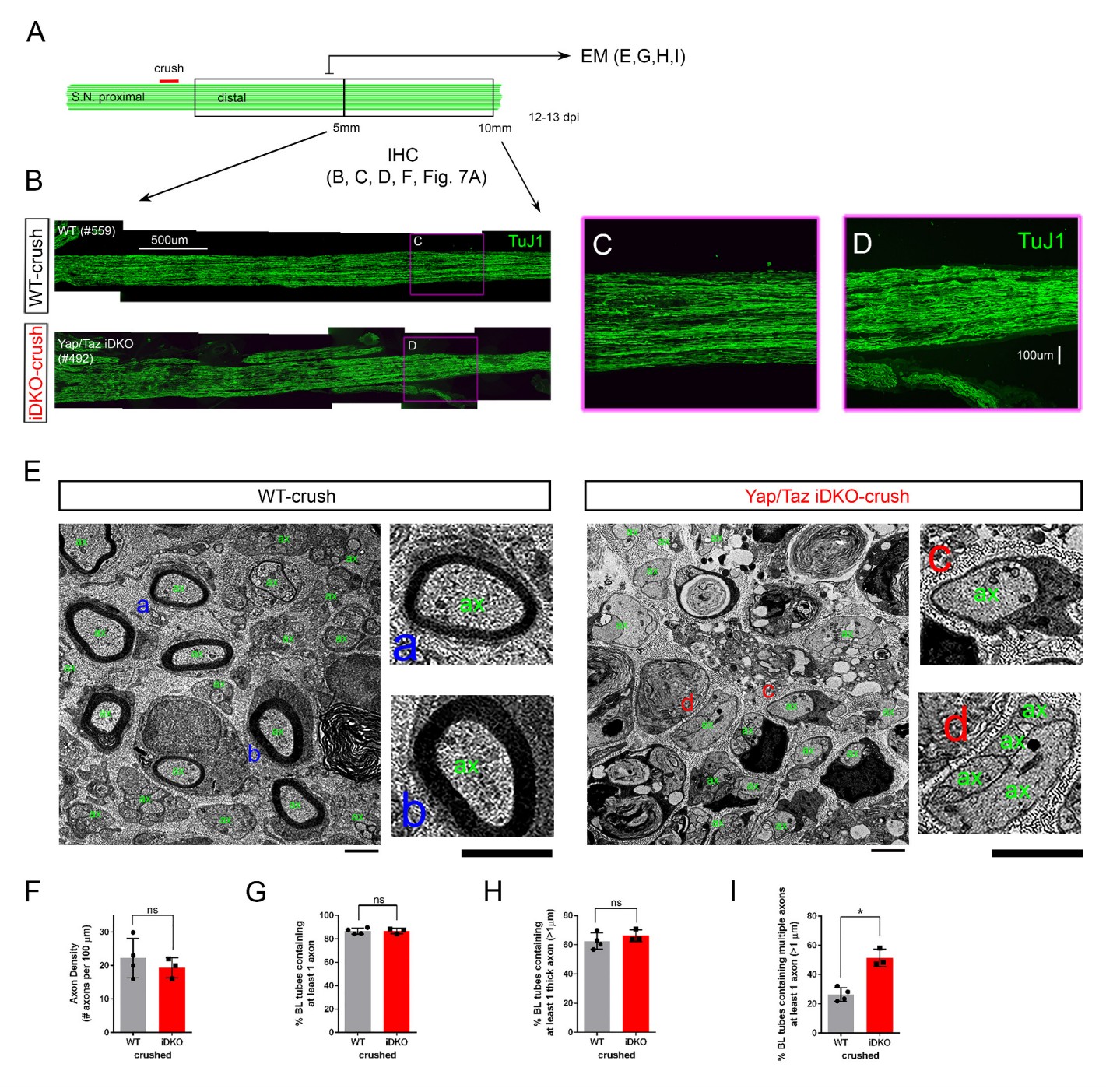

**Figure 5.** Schwann cells lacking YAP/TAZ support axon regeneration. (**A**) Schematic showing relative locations and sizes of the distal nerve segments used for ultrastructural or light microscopic analysis of axon regeneration in WT or Yap/Taz iDKO, 12–13 days after nerve crush. (**B**) Low magnification views of longitudinal sections of ~5 mm long nerve segments distal to the crush site, showing regenerated axons in iDKO as abundant as in WT. Axons are marked by TuJ1. (**C, D**) High magnification views of boxed area in (**B**), ~8 mm distal to the crush site. (**E**) Low and high magnification views of TEM, taken at 5 mm distal to the crush site, showing numerous axons that regenerated within basal lamina tubes in iDKO, as in WT. 'ax' denotes an axon. Numerous axons are large (>1 μm) but unmyelinated in iDKO. Examples of single large myelinated axons in WT (**E–a, E–b**), single large unmyelinated axon in iDKO (**E–c**) and axon bundles containing a large unmyelinated axon in iDKO (**E–d**). (**F**) Quantification of the axon density in crushed nerves of WT and iDKO, n = 4 mice for WT and three mice for iDKO. ns, not significant, p=0.4715, Mann-Whitney. (**G**) Quantification of the percentage of BL tubes containing axons in crushed nerves of WT and iDKO, n = 4 mice for WT and three mice for iDKO. ns, not significant, p=0.7237, Mann-Whitney (**H**) Quantification of the percentage of BL tubes containing at least one axon >1 μm in diameter, in crushed nerves of WT and iDKO. n = 4 mice for WT and three mice for iDKO. ns, not significant, p=0.1573, Mann-Whitney. (**I**) Quantification of the percentage of BL tubes containing multiple axons, at

*Figure 5 continued on next page*

*Figure 5 continued*

least one of which is >1 µm in diameter, in crushed nerves of WT and iDKO. n = 4 mice for WT and three mice for iDKO. *p=0.0339, Mann-Whitney.
Scale bars = 500 µm (**B**), 100 µm (**C**, **D**), 2 µm (**E**).
The online version of this article includes the following source data for figure 5:

**Source data 1.** Source files for TEM data.
**Source data 2.** Source files for graphs quantifying TEM data.

Deng et al. attributed the myelination failure primarily to the inability of SCs to differentiate into myelinating SCs (*Deng et al., 2017*; *Grove et al., 2017*). Deng et al., however, disagreed with our view about the role of YAP/TAZ in myelin maintenance of adult nerves (*Deng et al., 2017*; *Grove et al., 2017*). These disagreements motivated the present study, which investigated YAP/TAZ expression in adult SCs after nerve injury and their contribution to nerve regeneration. We found that YAP/TAZ dramatically disappear and reappear in SCs after nerve injury, and this loss and recovery of YAP/TAZ in SCs are spatiotemporally correlated with degeneration and regeneration of axons. We also found that SCs lacking YAP/TAZ proliferate and wrap around regenerated axons normally, but then fail to remyelinate them. These findings have several important implications for YAP/TAZ function in mature SCs.

Using antibodies specifically immunolabeling YAP or YAP/TAZ, we found dramatic down- and upregulation of both nuclear and cytoplasmic YAP/TAZ in SCs after nerve injury. Immunohistochemical identification of SC-selective YAP/TAZ was essential for detecting spatiotemporal regulation of YAP/TAZ. Indeed, we were only able to detect YAP/TAZ downregulation on Western blots when we used lysates prepared from nerves extensively perfused with saline, and from which the epi- and perineurium had been carefully removed. This procedure probably succeeded because it minimized the amount of YAP/TAZ present in cells other than SCs. Careful attention to YAP/TAZ expression in cells other than SCs will help to resolve inconsistencies in earlier studies of YAP/TAZ expression in peripheral nerve.

YAP/TAZ are located in the nuclei of developing SCs, where they promote proliferation and differentiation (*Poitelon et al., 2016*; *Deng et al., 2017*; *Grove et al., 2017*). They are also nuclear in adult SCs that maintain the myelin sheath (*Grove et al., 2017*) and that proliferate abnormally (*Wu et al., 2018*). It was therefore particularly intriguing to find that YAP/TAZ become undetectable in denervated SCs and that SCs lacking YAP/TAZ proliferate normally. YAP/TAZ disappeared from SCs, upon axon degeneration in both crushed and transected nerves. They reappeared in SCs in crushed nerve concomitant with regenerating axons, but not in transected nerve lacking axons, suggesting that YAP/TAZ expression in SCs is dependent on axons. It is also notable that YAP/TAZ appeared unchanged at one dpi, but had dramatically disappeared at three dpi, when axon degeneration was well underway (*Beirowski et al., 2005*; *Gomez-Sanchez et al., 2015*; *Jang et al., 2016*) and denervated SCs had lost contact with axons. Furthermore, SCs that upregulated YAP/TAZ after three dpi were associated with regenerating axons large enough to be myelinated. These results are in consistent with our earlier findings of selective expression of YAP/TAZ in myelin-forming SCs (*Grove et al., 2017*), implying that direct SC-axon contact probably regulate YAP/TAZ down- and upregulation after nerve injury.

We were surprised to find that SC proliferation proceeds normally in *Yap/Taz* iDKO nerves after injury. Proliferation of mature SCs after injury is therefore due to a YAP/TAZ-independent mechanism, in contrast to the proliferation of developing SCs, which is markedly reduced by YAP/TAZ inactivation (*Clemence et al., 1989*; *Grove et al., 2017*). This result is consistent with the notion that the mechanism for SC proliferation during development differs from that for proliferation after injury (*Atanasoski et al., 2001*; *Atanasoski et al., 2008*). However, our finding does not indicate that YAP/TAZ are unable to stimulate proliferation of mature SCs. Abnormally high levels of YAP have been shown to elicit excessive SC proliferation in Merlin mutants after nerve injury (*Mindos et al., 2017*), and YAP/TAZ overexpression induced by LATS1/2 inactivation has been shown to induce tumorigenic SC proliferation in adult nerves (*Wu et al., 2018*). These observations, together with our own, indicate that YAP/TAZ are not normally involved in injury-elicited SC proliferation, but that, if abnormally overexpressed, they can stimulate vigorous SC proliferation. It is also noteworthy that YAP/TAZ inactivation markedly reduces, but does not completely prevent, proliferation of

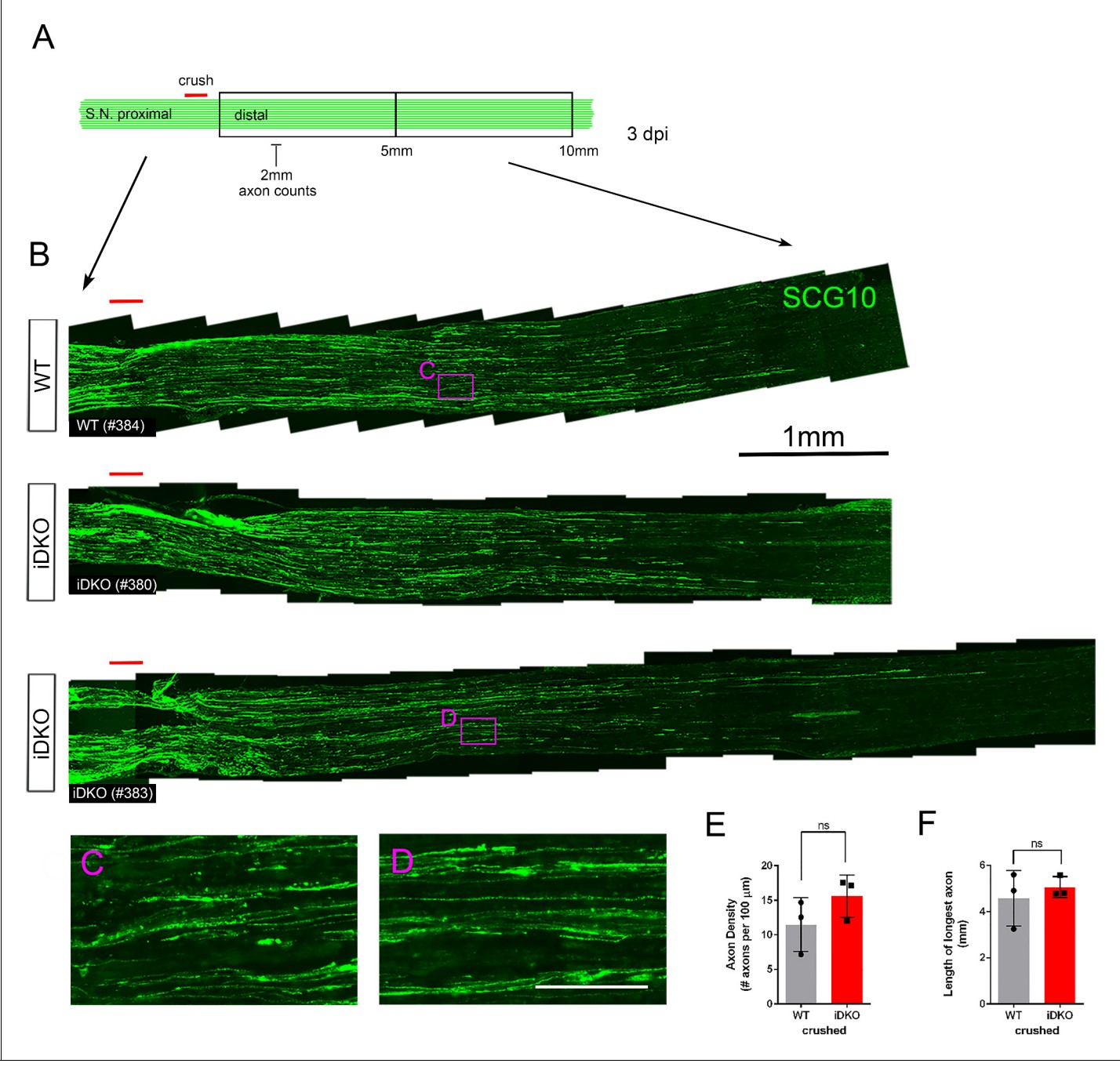

**Figure 6.** Schwann cells lacking YAP/TAZ support timely axon regeneration after acute injury. (A) Schematic showing relative locations of crushed site, axon quantification and sizes of the distal nerve segments used for light microscopic analysis of axon regeneration in WT or Yap/Taz iDKO, 3 days after nerve crush. (B) Low magnification views of longitudinal sections, showing abundant axon regeneration in both WT and iDKO. Regenerating axons are marked by SCG10. (C, D) High magnification views of boxed areas in (B), showing numerous thin regenerating axons. (E) Quantification of the axon density measured at 2 mm distal to the crushed site. n = 3 mice per genotype. ns, not significant, p=0.2752, Mann-Whitney. (F) Quantification of the distance regenerated by the longest axon. n = 3 mice per genotype. ns, not significant, p=0.8273, Mann-Whitney. Scale bars = 1 mm (B), 100 µm (C, D). The online version of this article includes the following source data for figure 6:

**Source data 1.** Source files for graphs quantifying axon density and length of longest axon.

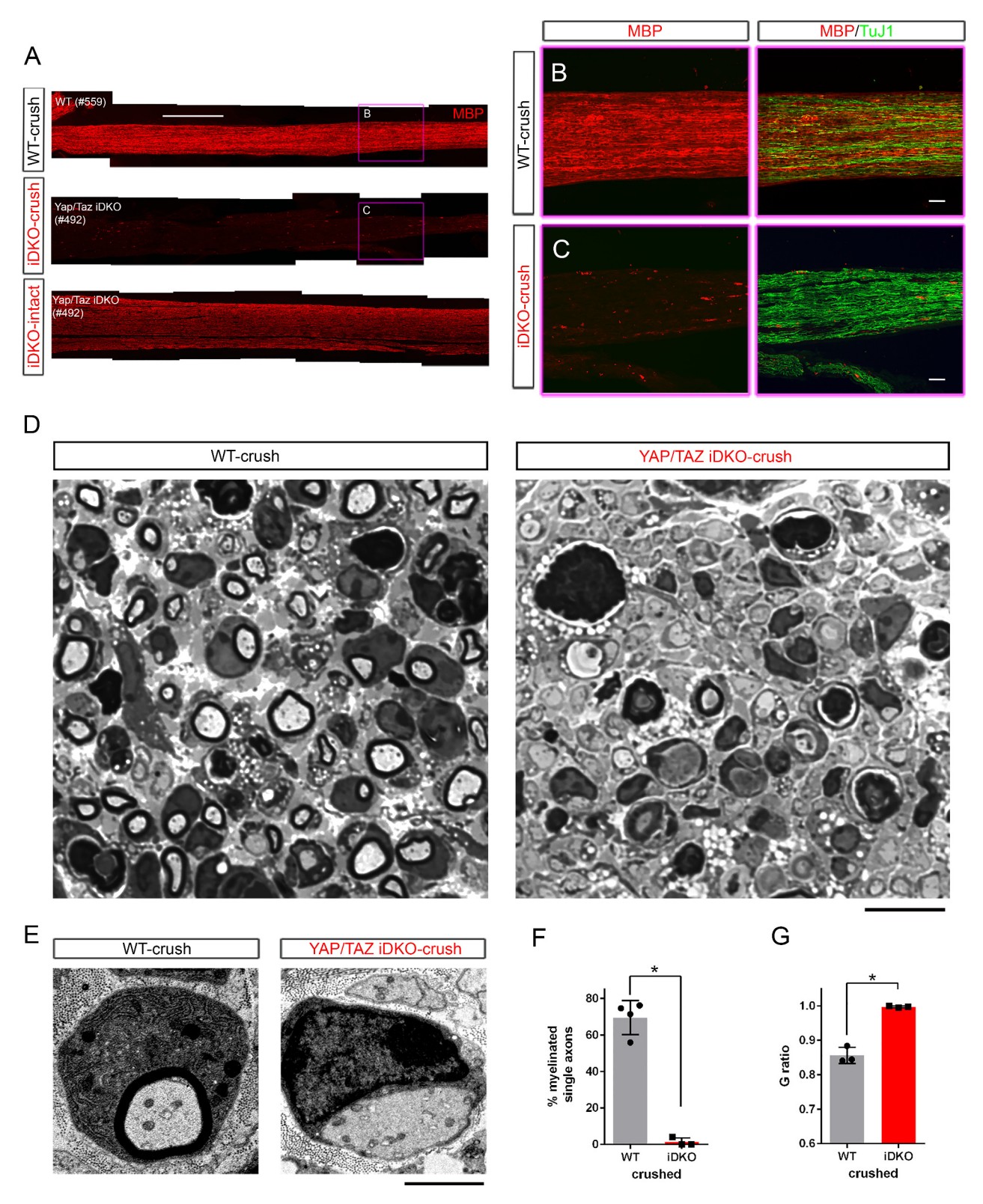

**Figure 7.** Schwann cells lacking YAP/TAZ fail to myelinate regenerated axons. Ultrastructural and light microscopic analyses of remyelination in distal nerves of WT or Yap/Taz iDKO, 12–13 days after nerve crush. (A) Low magnification views of longitudinal sections of intact or crushed nerves of WT and iDKO, showing no myelination of regenerated axons in crushed nerves of iDKO as indicated by the lack of MBP immunostaining. Refer to *Figure 5B* for robustly regenerated axons in the same iDKO mouse. (B, C) High magnification views of boxed area in (A), showing abundant regenerated axons in

*Figure 7 continued on next page*

*Figure 7 continued*

crushed nerves of both WT (**B**) and iDKO (**C**). Note that regenerated axons in iDKO are not myelinated. Axons and myelin are marked by TuJ1 and MBP, respectively. (**D**) Semi-thin sections stained with toluidine blue showing numerous myelinated axons in crushed nerves of WT but not in iDKO. (**E**) TEM images of representative single large axons, myelinated in WT (left panel) but unmyelinated in iDKO (right panel). (**F**) Quantification of the percentage of single axons that are myelinated. n = 4 mice for WT and three mice for iDKO. *p=0.0323, Mann-Whitney. (**G**) G-ratio in WT and iDKO. Myelinated axons in WT are compared to unmyelinated single axons in iDKO. n = 3 mice per genotype. *p=0.0495 Mann-Whitney. Scale bars = 500 μm (**A**), 100 μm (**B, C**), 10 μm (**D**), 2 μm (**E**).

The online version of this article includes the following source data and figure supplement(s) for figure 7:

**Source data 1.** Source files for TEM data.
**Source data 2.** Source files for graphs quantifying TEM data.
**Figure supplement 1.** Additional images of axon regeneration and remyelination in WT and *Yap/Taz* iDKO.

developing SCs (*Deng et al., 2017*; *Grove et al., 2017*). We suggest, therefore, that YAP/TAZ are potent stimulants of SC proliferation, but not an absolute requirement.

Tumorigenic proliferation of adult SCs associated with abnormally increased YAP/TAZ levels (*Wu et al., 2018*) suggests the importance of maintaining proper levels of YAP/TAZ, but it does not explain why YAP/TAZ are almost completely lost, rather than reduced, in denervated SCs. We previously demonstrated that inducible deletion of YAP/TAZ elicits SC demyelination in adult intact nerve (*Grove et al., 2017*). If YAP/TAZ indeed maintain myelination and act by promoting transcription of Krox20 and other myelin genes, then sustaining YAP/TAZ would counteract demyelination and dedifferentiation of SCs after injury. Conversely, their absence would promote downregulation of myelin genes, facilitating demyelination and formation of repair SCs. In accordance with these ideas, transcription of Krox 20 and other myelin genes remains robust in SCs up until two dpi, but is downregulated by three dpi (*Arthur-Farraj et al., 2017*), when we observed dramatic disappearance of YAP/TAZ. This timing suggests that YAP/TAZ protein downregulation leads to Krox 20 mRNA downregulation, suppressing expression of myelin proteins in de-differentiating Schwann cells. Complete loss of both nuclear and cytoplasmic YAP/TAZ could therefore imply that active regulatory mechanisms completely inactivate YAP/TAZ after injury. In support of this notion, in mutant mice lacking Merlin, YAP is abnormally upregulated in SCs after nerve injury, impairing SC de-differentiation (*Mindos et al., 2017*). This YAP upregulation suggests that one role of Merlin is to downregulate YAP in SCs and that YAP/TAZ expression in SCs is likely under active, presumably axon-dependent, regulation in both intact and injured mature nerves. This postulation of axon-dependent regulation of YAP/TAZ emphasizes that YAP/TAZ play a passive role in Wallerian degeneration, predicting that SCs do not require YAP/TAZ to dedifferentiate, proliferate, or transdifferentiate to repair SCs. Indeed, we found that these processes proceed normally in *Yap/Taz* iDKO mice. iDKO SCs are capable of wrapping around large diameter single axons but fail to initiate remyelination, which recapitulates the developmental phenotype of these mutant mice (*Deng et al., 2017*; *Grove et al., 2017*). For at least two reasons remyelination failure is highly unlikely to be due to poor physiological conditions of iDKO mice that die ~14 days after injury. First, axons regenerate normally in iDKO, which is unlikely if SCs are selectively vulnerable to poor physiological condition. Furthermore, iDKO SCs proliferate and trans-differentiate to repair-SCs normally. iDKO SCs also downregulate c-Jun to prepare for remyelination, whereas they maintain a higher level of Oct six than WT SCs, consistent with the failure of iDKO SCs to upregulate Krox 20 and MBP. Second, iDKO SCs wrap around individual axons, but fail to myelinate them, indicating that they proceed to the promyelination stage but no further. Therefore, one would have to postulate that the poor physiological condition of iDKO mice has a very specific effect on a particular remyelination stage, which we find unlikely.

iDKO SCs fail to upregulate Krox 20. Krox 20 is widely accepted as the key transcription factor promoting peripheral myelination. This is largely believed to be the case after injury (*Brügger et al., 2017*), although its role in remyelination has never been explicitly demonstrated. Other pathways can also promote upregulation of certain myelin proteins and lipids independently of Krox 20

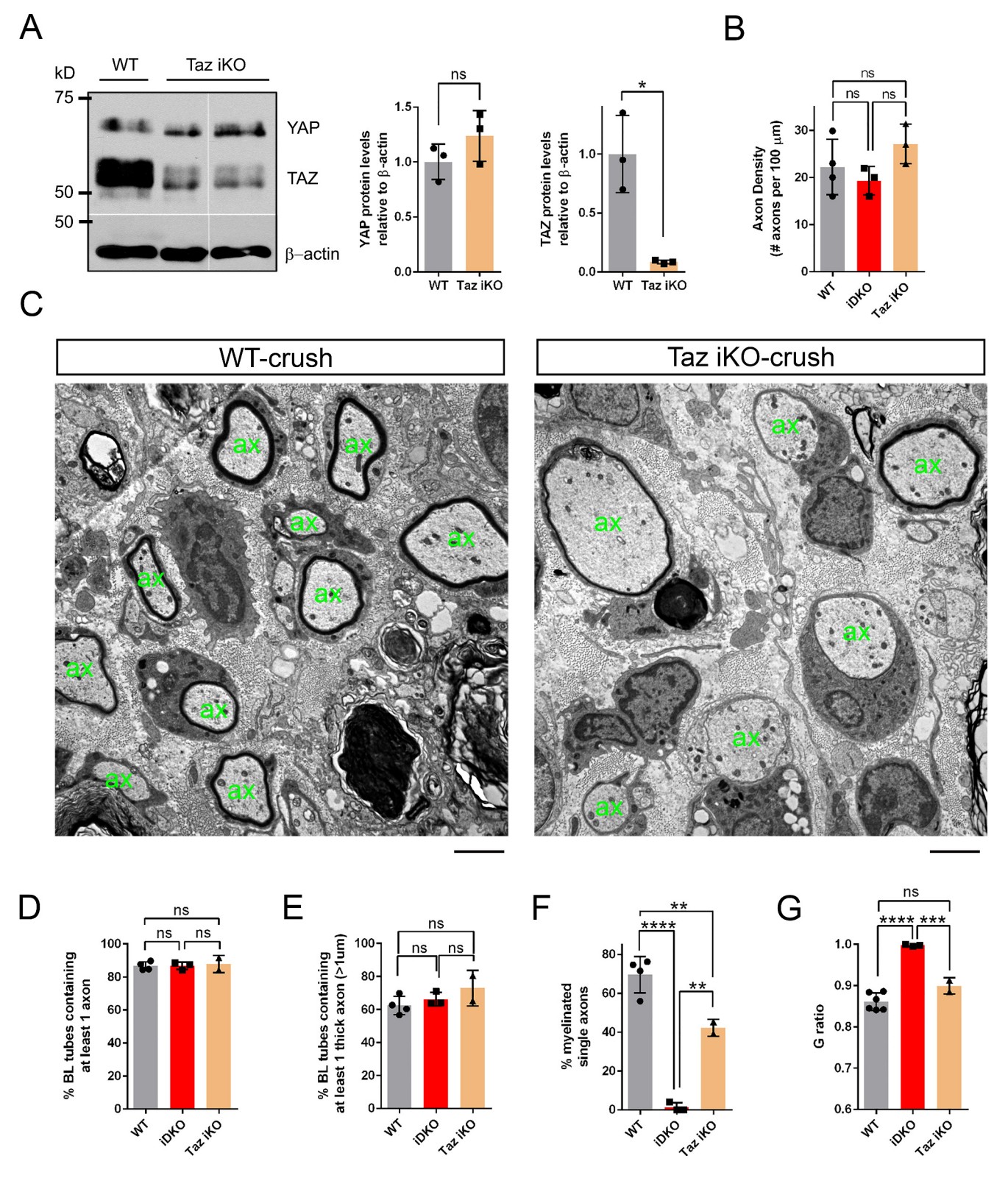

**Figure 8.** YAP and TAZ are redundantly required for optimal remyelination. (**A**) Western blotting of intact sciatic nerve lysates, showing markedly reduced TAZ in *Taz* iKO, whereas YAP levels remain relatively unchanged. YAP band is tighter and faster migrating in *Taz* iKO, than in WT, indicative of reduced phosphorylation. Quantification of Yap and Taz in WT and *Taz* iKO, n = 3 mice per genotype. YAP: ns, not significant, p=0.2752, Mann-Whitney. TAZ: *p=0.0495, Mann-Whitney. (**B**) Quantification of axon density in WT, *Yap/Taz* iDKO and *Taz* iKO nerves at 12 dpi, 8–10 mm distal to crush

*Figure 8 continued on next page*

*Figure 8 continued*

site (also see *Figure 5B,F* and *Figure 8—figure supplement 1B, (E)*). n = 4 mice for WT, three mice for iDKO and *Taz* iKO: WT vs iDKO, p=0.72; WT vs iKO, p=0.41; iDKO vs iKO, p=0.18, all not significant, one-way ANOVA with Tukey's multiple comparison test. (**C–G**) Comparative analysis of axon regeneration and remyelination in WT and *Taz* iKO, 12–13 days after nerve crush. (**C**) Representative TEM images of WT and *Taz* iKO nerves, taken at 5 mm distal to the crush site, showing numerous axons that regenerated within basal lamina tubes in *Taz* iKO, as in WT. 'ax' denotes a single axon. Some large axons are myelinated in *Taz* iKO. (**D**) Quantification of the percentage of BL tubes containing axons of any diameter in WT, *Taz* iKO and *Yap/Taz* iDKO nerves. n = 4 mice for WT, three mice for iDKO and two mice for Taz iKO: WT vs. iDKO, p=0.99; WT vs. iKO, p=0.90; iDKO vs. *Taz* iKO, p=0.92, all not significant, one-way ANOVA with Tukey's multiple comparison test. (**E**) Quantification of the percentage of BL tubes containing at least one axon larger than 1 μm in diameter in WT, *Taz* iKO and *Yap/Taz* iDKO nerves. n = 4 mice for WT, three mice for iDKO and two mice for *Taz* iKO: WT vs. iDKO, p=0.73; WT vs. iKO, p=0.22; iDKO vs. iKO, p=0.52, all not significant, one-way ANOVA with Tukey's multiple comparison test. (**F**) Quantification of the percentage of single axons that are remyelinated in WT, *Taz* iKO and *Yap/Taz* iDKO nerves. n = 4 mice for WT, three mice for iDKO and two mice for Taz iKO: WT vs. iDKO, ****p<0.0001; WT vs. iKO, **p=0.0094; iDKO vs. *Taz* iKO, **p=0.0016, one-way ANOVA with Tukey's multiple comparison test. (**G**) G-ratios of remyelinated axons in WT and *Taz* iKO nerves, compared to unmyelinated axons in *Yap/Taz* iDKO nerve. WT and *Taz* iKO remyelinated axons have equivalent G-ratios. n = 6 mice for WT, three mice for iDKO and two mice for iKO: WT vs. iDKO, ****p<0.0001; WT vs. iKO, not significant, p=0.074; iDKO vs. iKO, ***p=0.0008, one-way ANOVA with Tukey's multiple comparison test. Scale bar = 2 μm (**C**).

The online version of this article includes the following source data and figure supplement(s) for figure 8:

**Source data 1.** Source files for TEM data.
**Source data 2.** Source files for graphs quantifying Yap and Taz levels.
**Source data 3.** Source files for graphs quantifying axon density and TEM data.
**Source data 4.** Loss of TAZ protein expression in sciatic nerves of *Taz* iKO mice.
**Figure supplement 1.** Schwann cells expressing YAP (but lacking TAZ) support axon regeneration.
**Figure supplement 1—source data 1.** Source files for graph quantifying axon density.

(*Domènech-Estévez et al., 2016*), and numerous other factors, both positive and negative regulators, mediate peripheral myelination (*Jessen and Mirsky, 2008*; *Herbert and Monk, 2017*). Therefore, it is conceivable that the lack of Krox 20 in iDKO is a consequence rather than a cause of impaired remyelination. However, several considerations make it highly unlikely. First, we and others demonstrated that YAP/TAZ-TEAD1 complex directly binds to the cis-acting regulatory sequence of Krox 20 designated as the Myelinating Schwann cell Element (MSE) to upregulate Krox 20 during developmental myelination (*Lopez-Anido et al., 2016*; *Grove et al., 2017*). Second, we showed that YAP/TAZ-TEAD1 also bind to Krox 20 MSE in adult nerve, suggesting direct regulation of Krox 20 by YAP/TAZ in mature SCs (*Grove et al., 2017*). Third, Oct 6, which induces upregulation of Krox 20, together with other TFs, is upregulated in iDKO after nerve injury, as in WT (*Figures 4H* and *9G*). This finding suggests that remyelination in iDKO is blocked at the step of Krox 20 upregulation. Indeed, we and others have shown that iDKO SCs are arrested at the promyelination stage during development (*Deng et al., 2017*; *Grove et al., 2017*), and we found that, similarly after nerve injury, iDKO SCs proceed to the promyelination stage but fail to upregulate Krox 20 and initiate myelin formation.

YAP upregulation in SCs lacking Merlin has recently been reported to decrease the regeneration-promoting ability of repair SCs, which prevents axon regeneration in Merlin mutants (*Mindos et al., 2017*). This study implicates YAP as an inhibitor of axon regeneration. Our study suggests that this inhibition is dose- and context-dependent. We observed that repair SCs rapidly upregulate both YAP and TAZ as axons regenerate and that expression persists as regeneration continues. We also found that axon regeneration is as robust in *Taz* iKO and *Yap/Taz* iDKO as in WT, but not noticeably enhanced. Given that YAP is not compensatorily upregulated in *Taz* iKO (*Figure 8A*), these results suggest that at least normal levels of YAP do not prevent axon regeneration. However, overly robust upregulation of YAP, presumably as in Merlin mutants (*Mindos et al., 2017*), may severely compromise axon regeneration because excessive levels of YAP/TAZ alter the growth-promoting ability of SCs and/or cause their tumorigenic proliferation.

The present study, together with earlier work, strongly suggests that the levels of YAP/TAZ may be a critical determinant of their function in adult SCs. Optimal levels of YAP/TAZ promote myelin formation, maintenance and remyelination, whereas their absence promotes demyelination. In contrast, overly excessive levels of YAP/TAZ promote SC proliferation. Additional efforts to confirm this

notion and to understand the presumably axon-dependent mechanisms that tightly regulate nuclear levels, thus transcriptional activity, of YAP/TAZ in SCs may generate new strategies for peripheral nerve repair.

# Materials and methods

**Key resources table**

| Reagent type (species) or resource | Designation | Source or reference | Identifiers | Additional information |
|---|---|---|---|---|
| Strain, strain background (*Mus musculus*) | C57Bl/6 | Jackson Laboratory | Stock #: 000664; RRID:IMSR JAX:000664 | |
| Genetic reagent (*M. musculus*) | *Plp1*-Cre-ERT2 | | MGI:2663093 | (*Leone et al., 2003*) |
| Antibody | anti-Yap/Taz (rabbit monoclonal) | Cell Signaling Technology | D24E4, #8418 RRID:AB_10950494 | IHC 1:200 Western 1:1000 |
| Antibody | anti-SCG10 (rabbit monoclonal) | Novus Biologicals | NBP1-49461 RRID: AB_10011569 | IHC 1:5000 |
| Antibody | anti-Yap (rabbit monoclonal) | Cell Signaling Technology | D8H1X, #14074 RRID: AB_2650491 | IHC 1:200 |
| Antibody | anti-Sox10 (goat polyclonal) | R and D Systems | #AF-2864 RRID: AB_442208 | IHC 1:100 |
| Antibody | anti-Sox10 (rabbit monoclonal) | Abcam | EPR4007, #ab155279 RRID: AB_2650603 | IHC 1:250 |
| Antibody | anti-Egr2 (rabbit polyclonal) | Professor Dies Meijer, University of Edinburgh | | IHC 1:4000 |
| Antibody | anti-Oct6 (rabbit monoclonal) | Abcam | EP5421, #ab126746 RRID: AB_11130256 | WB 1:1000 |
| Antibody | anti-Oct6 (rabbit polyclonal) | Abcam | #ab31766 RRID: AB_776899 | IHC 1:800 |
| Antibody | anti-c-Jun (mouse monoclonal) | BD Transduction Laboratories | #610326 RRID: AB_397716 | IHC 1:500 |
| Antibody | anti-c-Jun (rabbit monoclonal) | Cell Signaling Technology | 60A8, #9165 RRID: AB_2130165 | WB 1:1000 |
| Antibody | anti-pS63-c-Jun (rabbit polyclonal) | Cell Signaling Technology | #9261 RRID: AB_2130162 | IHC 1:100 |
| Antibody | anti-Ki67 (rabbit polyclonal) | Abcam | #ab15580 RRID: AB_443209 | IHC 1:200 |
| Antibody | anti-p75NGFR (goat polyclonal) | Neuromics | #GT15057 RRID: AB_2737189 | IHC 1:400 |
| Antibody | anti-Tubulin β3 (rabbit polyclonal) | Biolegend | #802001 RRID: AB_2564645 | IHC 1:1000 |
| Antibody | IRDye-680 (goat anti-mouse) | LI-COR | #926–32220 RRID: AB_621840 | WB 1:15,000 |

*Continued on next page*

| Reagent type (species) or resource | Designation | Source or reference | Identifiers | Additional information |
|---|---|---|---|---|
| Antibody | HRP-Goat anti-mouse secondary antibody | Jackson Immunoresearch | #715-035-150 RRID: AB_2340770 | WB 1:12,000 |
| Antibody | HRP-Goat anti-rabbit secondary antibody | Jackson Immunoresearch | #115-055-062 RRID: AB_2338533 | WB 1:12,000 |
| Chemical compound, drug | Araldite 6005 | EMS | #10920 | |
| Chemical compound, drug | DDSA | EMS | #13710 | |
| Chemical compound, drug | DBP | EMS | #13101 | |
| Chemical compound, drug | BDMA | EMS | #11400–25 | |
| Other | Coated grids (100 mesh) | EMS | #FF100-Cu | |
| Chemical compound, drug | Osmium tetroxide (4% solution) | EMS | #19170 | |
| Chemical compound, drug | Lead nitrate | EMS | #17900 | |
| Chemical compound, drug | Sodium citrate | EMS | #21140 | |
| Chemical compound, drug | Uranyl acetate | EMS | #22400 | |
| Chemical compound, drug | Sodium borate | EMS | #21130 | |
| Chemical compound, drug | Toluidine blue | EMS | #22050 | |
| Chemical compound, drug | Paraformaldehyde | Sigma-Aldrich | #158127 | |
| Commercial assay or kit | Click-It EdU Alexa Fluor 594 kit | ThermoFisher Scientific | #C10339 | |
| Chemical compound, drug | EdU | ThermoFisher Scientific | #E10187 | |
| Chemical compound, drug | Tamoxifen | Sigma-Aldrich | #T5648 | |
| Other | DAPI stain | Invitrogen | #D1306 | IHC 1:250 |
| Antibody | Alexa 488, 568 or 647 secondaries | Jackson Immunoresearch | | IHC 1:250 to 1:1000 |
| Software, algorithm | Image Studio Lite | LI-COR, Inc | | |
| Software, algorithm | Prism | GraphPad Software, Inc | | |
| Software, algorithm | Stata | StataCorp LP | | Mann-Whitney test |

## Animals

All surgical procedures and animal maintenance complied with the National Institute of Health guidelines regarding the care and use of experimental animals and were approved by the Institutional Animal Care and Use Committee of Temple University, Philadelphia, PA, USA (Protocol 4920). Both male and female mice were used in all experiments, and were maintained on the C57BL/6 background. *Plp1-creERT2; Yap^{fl/fl}; Taz^{fl/fl}, Plp1-creERT2; Yap^{+/+}; Taz^{fl/fl}, Mpz-cre;Yap^{fl/fl}* and *Mpz-cre; Taz^{fl/fl}* mice used in this study were generated and genotyped as described previously (*Grove et al., 2017*). C57BL/6 mice were used for immunohistochemical analysis of YAP/TAZ.

## Tamoxifen administration

Tamoxifen was injected into 6–8 week old *Yap/Taz* iDKO or *Taz* iKO mice as previously described (*Grove and Brophy, 2014*). A 10 mg/ml solution of tamoxifen was made in 10:1 sunflower oil: 100% ethanol. This solution was injected intraperoneally at a concentration of 0.2 mg/g body weight. Injection was once daily for 5 days, followed by a 2 day break, then once daily for 5 consecutive days.

## Nerve crush or transection

Sciatic nerves of right hindlimbs were crushed or transected 24 hr after the final tamoxifen injection, using standard protocols (*Son and Thompson, 1995*). Briefly, a small skin incision was made in the posterior thigh and calf of the animals anesthetized by isoflurane. For crush, the sciatic nerve was crushed with a fine forceps (#5) for 10 s (3X) adjacent to the sciatic notch. The crush site was marked using charcoal-coated forceps, and the wound was closed. For transection, the exposed sciatic nerve was ligated at two directly adjacent sites, then cut with iridectomy scissors between the ligated sites. Ligated proximal and distal nerve endings were then sewn to adjacent muscle to prevent regeneration of axons from the proximal to distal nerve stumps. To identify proliferating Schwann cells, we intraperitoneally injected EdU (80 μg/g) eighty minutes before killing mice, as previously described (*Grove et al., 2017*).

## Western blotting

Mice were perfused with PBS, sciatic nerves removed, and epineurium and perineurium carefully stripped from the nerves. Western blotting followed the same procedure described previously (*Grove et al., 2017*), except for IRDye 680RD goat anti-mouse IgG (LiCor #926–68070; 1:5,000). Image Studio Lite (LI-COR Biosciences) was used for quantifying protein expression.

## Immunohistochemistry

Sciatic nerves were removed, and immediately fixed in 4% paraformaldehyde in PBS for 1 hr on ice. Nerves were washed 3 times in PBS, then stored in 15% sucrose in PBS overnight at 4°C for cryoprotection. Nerves were frozen-embedded in cryoprotectant medium (Thermo Fisher Scientific, Waltham, MA) in isomethylbutane at −80°C. 7–10 μm sections from the nerves were cut using a cryostat (Leica Microsystems, Germany) and collected directly onto glass slides. For immunolabeling, nerve sections were rehydrated in PBS, permeabilized in 0.5% Triton/PBS for 20 min, washed with PBS, then blocked in 2% bovine serum albumin (BSA) in PBS for 1 hr. Sections were incubated with primary antibodies in blocking buffer overnight at 4°C in a hydrated chamber, washed with PBS, and incubated with secondary antibodies in blocking buffer for 2 hr at room temperature. Sections were washed with PBS, stained with DAPI for 10 min, and mounted with Vectashield mounting medium (Vector Labs, Burlingame, CA). Nerve sections were incubated with antibodies previously described (*Grove et al., 2017*), except for the following: rabbit anti-Krox20 (kind gift from Professor Dies Meijer, Edinburgh, UK; 1:4000), rabbit anti-Yap (Cell Signaling #14074; 1:200), rabbit anti-SCG10 (Novus Biologicals #49461; 1:5000), goat anti-Sox10 (Santa Cruz #sc-17342; 1:200), goat anti-Sox10 (R and D Systems #AF-2864; 1:100), goat anti-p75 (Neuromics #GT15057; 1:400), rabbit anti-Ki67 (Abcam #ab15580; 1:1000), mouse anti-Tubulin β3 (clone Tuj1, Covance #MMS-435P; 1:1000), mouse anti-cJun (BD Biosciences #610326; 1:500), rabbit anti-cJun (CST #9165; 1:500), rabbit anti-phospho-cJun (CST #9261; 1:100).

## Electron microscopy, histology and morphometry

Sciatic nerves were removed and immediately fixed in EM buffer, as previously described (*Grove et al., 2017*). After nerve crush or transection, a 5 mm piece of the nerve was taken immediately distal to the injury site. The proximal end of the section was nicked with a razor blade for orientation during embedding. Fixation was for 2 hr at room temperature, followed by overnight at 4°C, with rotation. Post-fixation processing, embedding, cutting, staining and image capture were as previously described. For crushed or transected nerves, 500 nm semi-thin and 70 nm ultra-thin transverse sections were cut from the segment 5 mm distal to the crush/transection site.

For analysis of axon regeneration and remyelination, 7500x TEM sections were examined. This magnification allowed unambiguous identification of basal lamina tubes through which axons regenerate. Multiple non-overlapping images were taken for each section, such that all regions of each

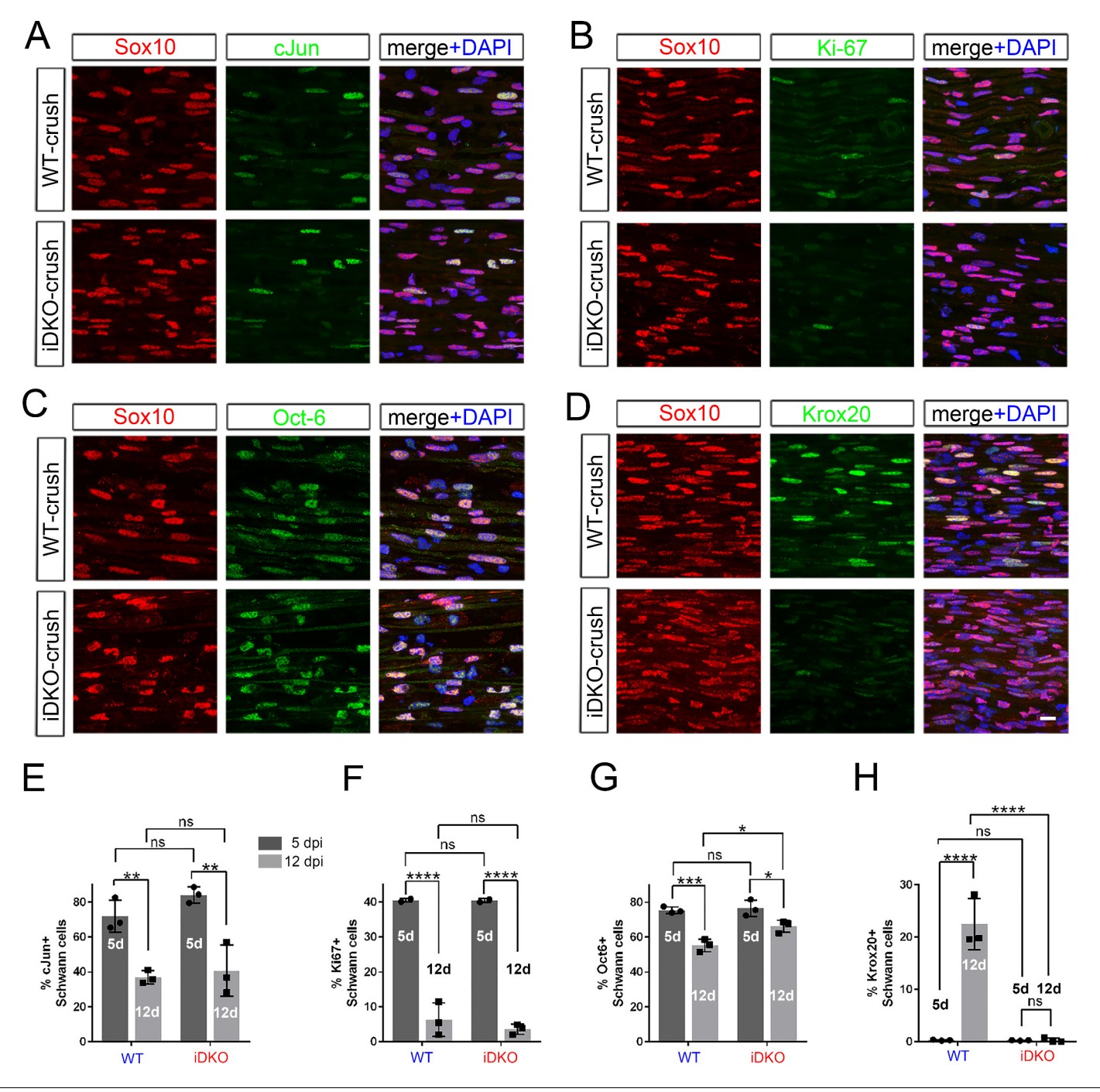

**Figure 9.** Redifferentiation of Schwann cells lacking YAP/TAZ. Longitudinal sections of crushed nerves of WT and *Yap/Taz* iDKO at 12 dpi, immunostained by various markers of SC dedifferentiation (c-Jun and Oct-6), proliferation (Ki67) and redifferentiation (Krox20). SCs are marked by Sox10. (**A**) Representative sections showing c-Jun+ SCs markedly reduced in iDKO, as in WT. (**B**) Representative sections showing rarely observed Ki67+ proliferating SCs in iDKO, as in WT. (**C**) Representative sections showing Oct-6+ SCs reduced in iDKO, as in WT. (**D**) Representative sections showing failed upregulation of Krox20 in iDKO SCs. (**E**) Quantitative comparison of c-Jun+ SCs at 5 and 12 dpi, showing similar downregulation of c-Jun in WT and iDKO SC. n = 3 mice per genotype, 2-way ANOVA, ns = not significant. WT five dpi vs WT 12 dpi, **p=0.0069; WT five dpi vs iDKO five dpi, p=0.4260; WT 12 dpi vs iDKO 12 dpi, p=0.9574; iDKO five dpi vs iDKO 12 dpi, **p=0.0018. (**F**) Quantitative comparison of Ki67+ SCs, showing similar reduction in proliferating SCs in WT and iDKO nerves between 5 dpi and 12 dpi. n = 3 mice per genotype, 2-way ANOVA, ns = not significant. WT five dpi vs WT 12 dpi, ****p<0.0001; WT five dpi vs iDKO five dpi, p>0.9999; WT 12 dpi vs iDKO 12 dpi, p=0.6775; iDKO five dpi vs iDKO 12 dpi, ****p<0.0001. (**G**) Quantitative comparison of Oct-6+ SCs, showing significant downregulation of Oct-6 in WT and iDKO SCs between 5 dpi and 12 dpi. n = 3 mice per genotype, ns = not significant, 2-way ANOVA. WT five dpi vs WT 12 dpi, ***p=0.0005; WT five dpi vs iDKO five dpi, p=0.9817; WT 12

*Figure 9 continued on next page*

*Figure 9 continued*

dpi vs iDKO 12 dpi, *p=0.0221; iDKO five dpi vs iDKO 12 dpi, *p=0.0299. (**H**) Quantitative comparison of Krox20+ SCs, showing upregulation of Krox20 in WT SCs, but not in iDKO SCs between 5 dpi and 12 dpi. n = 3 mice per genotype, 2-way ANOVA, ns = not significant. WT five dpi vs WT 12 dpi, ****p<0.0001; WT five dpi vs iDKO five dpi, p>0.9999; WT 12 dpi vs iDKO 12 dpi, ****p<0.0001; iDKO five dpi vs iDKO 12 dpi, p>0.9999. Scale bar = 10 µm (**A–D**).

The online version of this article includes the following source data for figure 9:

**Source data 1.** Source files for Krox20$^+$ SC data.
**Source data 2.** Source files for graphs quantifying c-Jun+ SCs, Ki67+ SCs, Oct6+ SCs and Krox20+ SCs.

section were sampled. Image J was used for image analysis. After counting the total number of basal lamina (BL) tubes per image, we next counted the number of BL tubes in the following categories: contains no axon(s); contains axon(s); contains at least one axon >1 µm in diameter; contains a single axon >1 µm in diameter; contains a myelinated axon. This procedure enabled us to calculate the percentage of BL tubes in each category. Using an ImageJ G-ratio calculator plug-in, G ratios for each genotype were calculated in two different ways: (1) All single large axons were counted, whether or not they were myelinated; (2) Only myelinated axons were counted.

## Data analysis

In each experiment, data collection and analysis were performed identically, regardless of mouse genotype. Data are presented as mean + / - SD. Statistical analysis was done using the two-sample Mann-Whitney test for two-group comparisons and analysis of variance (ANOVA) with Tukey's test for multiple comparisons, according to the number of samples and the analysis of mice at multiple ages. Sample sizes were similar to those employed in the field and are indicated in the main text, methods or figure legends. A p-value of 0.05 or less was considered statistically significant.

## Acknowledgements

We thank Alan Tessler and members of the Son laboratory for critical reading of the manuscript. We thank Dr. Hyukmin Kim for intraperitoneal injection, Dr. Eric Olson for Yap and Taz floxed mice, Drs. Ueli Suter and Kelly Monk for Plp-creERT2 mice. Plp-creERT2 mice were generated by Dr. Ueli Suter using a patented Cre-ERT2 construct developed by Dr. Pierre Chambon at GIE-CERBM. This paper is dedicated to the memory of Dr. Wesley Thompson who revealed the pivotal roles of terminal Schwann cells in forming and restoring nerve-muscle connection.

## Additional information

### Funding

| Funder | Grant reference number | Author |
|---|---|---|
| National Institute of Neurological Disorders and Stroke | NS105796 | Young-Jin Son |
| Shriners Hospitals for Children | Research award | Young-Jin Son |

The funders had no role in study design, data collection and interpretation, or the decision to submit the work for publication.

### Author contributions

Matthew Grove, Conceptualization, Data curation, Formal analysis, Investigation; Hyunkyoung Lee, Data curation, Investigation; Huaqing Zhao, Statistical analysis; Young-Jin Son, Conceptualization, Data curation, Writing - original draft, Writing - review and editing

### Author ORCIDs

Young-Jin Son  https://orcid.org/0000-0001-5725-9775

## Ethics

Animal experimentation: All surgical procedures and animal maintenance complied with the National Institute of Health guidelines regarding the care and use of experimental animals and were approved by the Institutional Animal Care and Use Committee of Temple University, Philadelphia, PA, USA. Protocol 4920.

## Decision letter and Author response

Decision letter https://doi.org/10.7554/eLife.50138.sa1
Author response https://doi.org/10.7554/eLife.50138.sa2

## Additional files

### Supplementary files

• Transparent reporting form

### Data availability

All data generated during this study are included in the manuscript.

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
