## [Decision Letter]

**Acceptance summary:**

This follow-up study on the authors' previous work examines the effects of transcriptional activity of YAP/TAZ in Schwann cell response to peripheral nerve injury. Using in vivo analyses of mutant mice, the authors show that YAP/TAZ function is not essential for Schwann cell proliferation and transdifferentiation into repair cells but is required for remyelination. This comprehensive work advances our understanding of regeneration of peripheral nervous system.

**Decision letter after peer review:**

Thank you for submitting your article "Axon-dependent expression of YAP/TAZ mediates Schwann cell remyelination but not proliferation after nerve injury" for consideration by *eLife*. Your article has been reviewed by three peer reviewers, and the evaluation has been overseen by a Reviewing Editor and Anna Akhmanova as the Senior Editor. The following individuals involved in review of your submission have agreed to reveal their identity: Michael Wehr (Reviewer #2).

The reviewers have discussed the reviews with one another and the Reviewing Editor has drafted this decision to help you prepare a revised submission.

The reviewers liked your work, in which you sought to determine the role of YAP/TAZ after nerve injury. This work is a follow-up of an earlier paper in which the team had assessed the role of these transcription factors in developing Schwann cells and myelination.

While YAP/TAZ deletion in Schwann cells does not affect Schwann cells trans-differentiation into repair Schwann cells (nor axonal regeneration) following PNS injury, the authors show here that YAP and TAZ control the extent of remyelination after injury. YAP and TAZ disappear in denervated Schwann cells, but re-appear in these cells when axons regenerate. In addition, the authors report an effect on Krox 20 expression in mutant mice but not of several other factors, such as c-jun, p75NTR, Oct6. Mutant nerves fail to form basal lamina bundles during regeneration. The findings are interesting and potentially significant to the field of PNS injury

The work has been thoroughly performed and the data presented support the overall assessment. Nevertheless, there are a number of issues that need to be addressed.

Major points:

1) Figure 1

Figure 1B: It is not clear whether denervation results in loss of YAP/TAZ expression or prevents the nuclear localization (thus, the transcriptionally activity) in Schwann cells. If it is the latter, YAP/TAZ is expected to translocate to the cytoplasm following injury, which does not seem to be the case. Accordingly, the authors state that both nuclear and cytoplasmic YAP/TAZ are down regulated in Schwann cells. However, the sentence "SCs are critically dependent on axons for the transcriptional activity of YAP/TAZ" could be misleading and should be rephrased.

Figure 1D: the authors mention that YAP is weakly expressed 24dpi, nevertheless the extent of expression is comparable to that observed at 6dpi. Please rephrase as this assay is not quantitative.

Figure 1 E and F show that disappearance and re-appearance of YAP/TAZ correlates with the timing of axonal degeneration and regeneration, respectively. However, the authors do not show that YAP/TAZ expression depends on the axon. To make the claim, one should look for direct evidence and causality.

2) The conclusion that YAP expression critically depends on axons (subsection “YAP/TAZ expression in Schwann cells is axon-dependent”) appears too strong as at 6dpi axons are starting to regenerate. This concept is also extensively presented in the Discussion where it is claimed that YAP nuclear expression "is highly dependent on axons" (Discussion paragraph three). While this is likely true 24 dpi, at 6 dpi the axons are still (actively) degenerating.

3) One of the main observations in iDKO is the defect in basal lamina (BL) tube formation. Can one exclude that this defect is due to a different deposition of BL components between WT and iDKO mice? Can the authors perform a western blotting analyses to determine its composition?.

4) The reduction in Krox 20 expression could alternatively be a consequence rather than the cause of impaired remyelination. Please discuss and soften conclusions.

5) In their discussion the authors posit that the disappearance of YAP/TAZ from denervated nerves provides support for a role in myelin maintenance, unfortunately without data corroborating this. It is also possible that YAP/TAZ dowregulation is merely due to the ongoing de-differentiation occurring in injured Schwann cells, which is an essential pre-requisite to acquire a repair phenotype (Discussion paragraph one).

6) Based on the lack of expression of YAP/TAZ at 3dpi, one cannot conclude SC-axon interactions are essential to maintain YAP/TAZ expression in adult SCs. Denervated Schwann cells have a completely different phenotype from those observed in adult nerves (Discussion paragraph three).

7) In Discussion paragraph four, the authors propose that "YAP/TAZ powerfully stimulate developing and adult SCs to proliferate at all stages". This conclusion is not directly supported by data. The authors point out that in a recent study (Mindos et al., 2017) YAP and TAZ are critically regulated by Merlin in Schwann cells (SCs), and inactivation of Merlin leads to higher levels of YAP and TAZ preventing axon regeneration and remyelination. However, it remains unclear how YAP (in Taz iKO nerves) can myelinate regenerated axons. Would removal of Merlin in Taz iKO nerves also prevent myelination? An experiment that identifies these mechanisms would help.

8) The authors should check the statistics they applied. A t-test assumes that samples are normally distributed. This is not likely the case for 3 replicates. Use the Mann-Whitney test as alternative. Western blots should also be quantified and for all quantifications (imaging, Western blotting) individual data points should be shown.

9) Figure 4: Does the loss of YAP/TAZ result in a delay of axonal re-generation? Mindos et al., 2017, had shown in mice hat regenerating axons could reach up to 13-14 mm distal to the injury site within 7 days of nerve crush. The present study accessed regeneration 12-14 days after injury but only within the 5-10 mm distal region, a paradigm unlikely to detect a delay in axonal regeneration if present. To address this issue, regeneration should be analyzed earlier and ideally along its time course.

10) Similarly Figures 6, 7: Nerve regeneration/remyelination was accessed 12-13 dpi, a couple of days before death of the mice is expected. As stated in subsection “SCs lacking YAP/TAZ convert to repair SCs and support axon regeneration”, iDKO mice die about 14 days after tamoxifen injection (also stated in Grove et al., 2017). At the time point analyzed, even uninjured iDKO mice exhibit clear PNS nerve defects (shivering, decrease in nerve conduction velocity, demyelination etc) as well as ataxia and impaired breathing (Grove et al., 2017). This complicates interpretation of the data on remyelination as the failure to re-build myelin could be due to a secondary effect stemming from the poor physiological condition of the mice. To prove that the remyelination failure is indeed due to the loss of YAP/TAZ, one would like to see a rescue experiment that restores the YAP/TAZ function in the iDKO Schwann cells. Unless such experiments are provided, the conclusion and discussion must be carefully rephrased.

[Editors' note: further revisions were suggested prior to acceptance, as described below.]

Thank you for resubmitting your work entitled "Axon-dependent expression of YAP/TAZ mediates Schwann cell remyelination but not proliferation after nerve injury" for further consideration by *eLife*. Your revised article has been evaluated by Anna Akhmanova (Senior Editor) and a Reviewing Editor.

The manuscript has been improved but there are some remaining issues that need to be addressed before acceptance, as outlined below. The reviewers agreed that their remaining concerns can be addressed without performing additional experiments, by textual changes and clarifications and by providing source data.

Reviewer #1:

The revised version of the manuscript has significantly improved as compared to the original study. Nevertheless, the authors have not performed the requested additional experiments, which are instead argued in the rebuttal letter.

While it is clear from the presented data that p-YAP relocalizes in cytoplasm (Figure 2D), in Figure 1B the authors show and state that " the dramatic down- and up- regulation of YAP/TAZ concurrent with axon degeneration and regeneration, respectively, suggest that SCs are dependent on axons from YAP/TAZ nuclear localization". Given the experimental data it would be more correct talking about expression rather than localization. The latter indeed assumes a shuttle between in an out of the nucleus, which is valid for p-YAP and also in that case only partially.

Reviewer #2:

Grove et al. provide a detailed point-to-point response that satisfactorily discusses all issues raised.

However, this reviewer still has the following concerns related to the quantification of WB data (see below). Once clarified, the manuscript can be granted for publication.

Figure 1C, Figure 1—figure supplement 2:

For the quantification, 4 data points are shown per condition. Supplemental data show 2 uncropped blots that are 2 different exposures from the very same gel as presented in 1C (as the authors also specify). However, 3 independently performed blots that should show independently processed samples and were used for the quantification are not available. The authors need to provide the correct source data. Also note that Figure 1—figure supplement 2 contains a staining of nerve fibers, but no WB data.

Figure 4—figure supplement 1 and 2:

The quantification implicates two blots that were quantified. However, supplement 2 contains the same blot as presented in supplement 1. The authors need to provide the correct source data.

Figure 8A, Figure 8—figure supplement 2:

The quantification implicates 3 blots that were quantified. However, supplement 2 contains the same blot as presented in Figure 8A. Again, the authors need to provide the correct source data for all blots quantified.

Reviewer #3:

This is a revised manuscript which investigates the role of YAP/TAZ in mediating Schwann cell response to injury and PNS regeneration. The authors have addressed reviewers concerns sufficiently, by either conducting additional experiments or in writing. Overall, data from the study are solid and the findings are potentially significant to the field of PNS injury and regeneration. I only have one concern, which can be addressed in writing by the authors.

Figure 1 and Figure 1—figure supplement 1 (paragraph four subsection “YAP/TAZ expression in Schwann cells is axon-dependent”). I am still confused about the discrepancy between the WB and IHC results on YAP expression in denervated nerves. When detected WB, YAP expression continues to decrease after nerve crush and does not recover, whereas when assessed by IHC, YAP expression returns at 6 and 12D. The authors claim that the WB result reflects YAP expression in other cell types (I have to say, I do not agree with the explanation). However, they also make a point in the Discussion that WB was done using lysates prepared from nerves that had been de-sheathed (epi- and perineurium removed) to ensure enrichment of SC-derived proteins.

Since most of the YAP is nuclear at 6 and 12D, shown by the IHC, is it possible that the lysis buffer used for WB does not recover nuclear YAP in the Schwann cells, thus the WB only reflects the cytoplasmic YAP?

---

## [Author Response]

Major points:1) Figure 1Figure 1B: It is not clear whether denervation results in loss of YAP/TAZ expression or prevents the nuclear localization (thus, the transcriptionally activity) in Schwann cells. If it is the latter, YAP/TAZ is expected to translocate to the cytoplasm following injury, which does not seem to be the case. Accordingly, the authors state that both nuclear and cytoplasmic YAP/TAZ are down regulated in Schwann cells. However, the sentence "SCs are critically dependent on axons for the transcriptional activity of YAP/TAZ" could be misleading and should be rephrased.Figure 1D: the authors mention that YAP is weakly expressed 24dpi, nevertheless the extent of expression is comparable to that observed at 6dpi. Please rephrase as this assay is not quantitative.Figure 1 E and F show that disappearance and re-appearance of YAP/TAZ correlates with the timing of axonal degeneration and regeneration, respectively. However, the authors do not show that YAP/TAZ expression depends on the axon. To make the claim, one should look for direct evidence and causality.

Figure 1B (now Figure 2B): We have added Figure 2D, which shows that cytoplasmic expression of YAP (i.e., pYAP) returns in Schwann cells (SCs) of the crushed nerve, but not in those of the transected nerve in which axon regeneration into the distal stump is prevented (Figure 2C). This data are additional evidence that axons are necessary for the recovery of cytoplasmic expression of YAP/TAZ in SCs. We have also amended our statements on axon-dependence of YAP/TAZ expression.

Figure 1—figure supplement 1D (now Figure 1—figure supplement 1B): We have removed the confusing statement and just mention here that “YAP is upregulated in many SC nuclei at or after 6 dpi…”.

Figure 1E and F (now Figure 1B and C): Please note that we do NOT claim that SCs depend on “axon contact” for YAP/TAZ expression. We believe that this is likely to be the case (see below #2 and #6 responses) but we agree with the reviewer that only correlative data support it. We claim that they depend on “axons” (i.e., presence of axons implicating neural regulation) mainly based on the nerve transection experiment (previous Figure 1B and C, now Figure 2). Whereas crushing nerve permits regeneration of axons into the distal nerve stump, completely cutting and tying nerve (as we did in the nerve transection experiment) prevents axons from regenerating into distal nerve stump. As the distal stump of transected/tied nerves will not have axons, comparing protein expression in crushed versus transected/tied nerve has been used to examine axonal dependence or neural regulation in vivo. Our finding that YAP/TAZ disappear in SCs upon axon degeneration in both crushed and transected nerves, that they reappear in crushed nerve concomitant with regenerating axons, but not in transected nerve lacking axons therefore provides strong evidence for the dependence of YAP/TAZ expression in SCs on axons. Whether this neural regulation is via direct axon-SC contact or releasable factors or is mediated by another cell type awaits additional study.

2) The conclusion that YAP expression critically depends on axons (subsection “YAP/TAZ expression in Schwann cells is axon-dependent”) appears too strong as at 6dpi axons are starting to regenerate. This concept is also extensively presented in the Discussion where it is claimed that YAP nuclear expression "is highly dependent on axons" (Discussion paragraph three). While this is likely true 24 dpi, at 6 dpi the axons are still (actively) degenerating.

It is true that axon degeneration can be active at 6 dpi or even later. It is also important to note that, in the nerve crush injury model, new axons form and penetrate the crush site within 1-2 days after injury and extend further along the distal nerve stump. As shown in the newly provided Figure 6 and in many other studies, regenerating axons can reach up to ~4 mm distal to the crush site at 3 dpi. These axons then keep regenerating within the basal lamina (BL) tubes at the speed of 1-4 mm/day, although the debris of degenerating axons and myelin are not yet completely removed. We observed regenerating axons at 3 dpi (e.g., Figure 1B; zoomed area of 3D-dstl) but many more (and thicker, large-diameter) axons at 6 dpi which made contacts with SCs (Figure 1B; zoomed area of 6D-dstl). These data therefore support our conclusion and further suggest that regenerating axons upregulate YAP/TAZ in SCs. However, we agree with the reviewer that our statements were too strong, and we have modulated the sentences.

3) One of the main observations in iDKO is the defect in basal lamina (BL) tube formation. Can one exclude that this defect is due to a different deposition of BL components between WT and iDKO mice? Can the authors perform a western blotting analyses to determine its composition?.

We assume that the reviewer meant failed remyelination by “…defect in BL formation…”, and that s/he wondered if failed remyelination in iDKO could be due to defective signaling from BL evoked by the loss of YAP/TAZ in SCs. First, it is our understanding that BL is important but axons and SCs also provide essential components for optimal remyelination. The complete, rather than partial or delayed, loss of remyelination that we observed in iDKO is therefore likely to be due to inability of SCs in iDKO to coordinate signals from axons and SCs, as well as those from BL. Second, we consider it unlikely that substantial changes can be made in BL tubes in adult iDKO mice, as they are already established and completely normal before tamoxifen injection. Considering the marked structural and molecular stability of BL tubes in mature nerves, we anticipate modest if any changes in BL composition during the relatively short time between tamoxifen injection and terminal end points. In support of this notion, axons regenerate normally within BL tubes in iDKO (Figure 5). Axon regeneration should have been attenuated if there were substantial changes in BL associated growth-promoting molecules such as laminin. Laminin is a main BL component also known to promote myelination. Thus, we believe it reasonable to predict that SCs fail to remyelinate although a substantial amount of laminin is present in BL of iDKO.

4) The reduction in Krox 20 expression could alternatively be a consequence rather than the cause of impaired remyelination. Please discuss and soften conclusions.

We appreciate the comment and have amended our statements and discussed as suggested. Krox 20 is widely accepted as the key transcription factor (TF) promoting peripheral myelination. This is largely believed to be the case after injury, although its role in remyelination has never been explicitly demonstrated. Other pathways can also promote upregulation of certain myelin proteins and lipids independently of Krox 20, and numerous other factors, both positive and negative regulators, mediate peripheral myelination. Therefore, it is conceivable that the lack of Krox 20 in iDKO is a consequence rather than a cause of impaired myelination. However, several considerations make it highly unlikely. First, we and others have shown that YAP/TAZ-TEAD1 complex directly binds to the cis-acting regulatory sequence of Krox 20 (i.e., MSE) to upregulate Krox 20 during developmental myelination. Second, we showed that YAP/TAZ-TEAD1 also bind to Krox 20 MSE in mature nerve, suggesting direct regulation of Krox 20 by YAP/TAZ in mature SCs. Third, Oct 6, which induces upregulation of Krox 20, together with other TFs, is upregulated in iDKO after nerve injury, as in WT (Figure 4H, 9G). This finding suggests that remyelination in iDKO is blocked at the step of Krox 20 upregulation. Indeed, we and others have shown that iDKO SCs are arrested at the promyelination stage during development, and similarly after nerve injury, we found that iDKO SCs are capable of wrapping around individual axons but fail to upregulate Krox 20 and initiate myelin formation (Figure 5E, 7E).

5) In their discussion the authors posit that the disappearance of YAP/TAZ from denervated nerves provides support for a role in myelin maintenance, unfortunately without data corroborating this. It is also possible that YAP/TAZ dowregulation is merely due to the ongoing de-differentiation occurring in injured Schwann cells, which is an essential pre-requisite to acquire a repair phenotype (Discussion paragraph one).

We reduced and clarified our discussion of the “implication” of the present remyelination study for the role of YAP/TAZ in myelin maintenance. Due to conflicting reports among different groups, it remains controversial whether YAP/TAZ are crucial also for maintaining myelin in mature nerve. This is a very important issue in the field, and intimately associated with demyelination and remyelination of mature nerve after injury. Therefore, we believe that our discussion of the “implication” of the present study is sufficiently justified and does not require direct supporting data. We also discuss prior studies that support the notion that YAP/TAZ downregulation may be actively regulated, rather than passive due to ongoing de-differentiation. These findings include that Krox 20 mRNA is not downregulated until 3 days after injury (Arthur-Farraj et al., 2017), when we observed YAP/TAZ protein downregulation. This timing suggests that YAP/TAZ protein downregulation leads to Krox 20 mRNA downregulation, suppressing expression of myelin proteins in de-differentiating SCs. Complete loss of both nuclear and cytoplasmic YAP/TAZ could therefore implicate possible presence of active regulatory mechanisms that completely inactivate YAP/TAZ after injury. In support of this notion, in mutant mice lacking Merlin, YAP is abnormally upregulated in SCs after nerve injury, impairing SC de-differentiation (Mindos et al., 2017). This YAP upregulation suggests that one role of Merlin is to downregulate YAP in SCs after injury and that YAP/TAZ expression in SCs is likely under active regulation in both intact and injured mature nerves.

6) Based on the lack of expression of YAP/TAZ at 3dpi, one cannot conclude SC-axon interactions are essential to maintain YAP/TAZ expression in adult SCs. Denervated Schwann cells have a completely different phenotype from those observed in adult nerves (Discussion paragraph three).

Here again we meant to discuss the implications of the remyelination data for myelin maintenance, not as a conclusion. We have now softened and clarified the statement so that readers do not misinterpret us. As mentioned in our response to #2 comment, we claim that SCs depend on axons for YAP/TAZ expression after injury. We now provide images of an enlarged area of crushed and transected/tied nerves (Figure 1B, Figure 2B, 2C, 2D) to better demonstrate that loss and recovery of YAP/TAP expression in SCs are spatiotemporally correlated with loss and recovery of SC-axon contact. It is particularly interesting that a few thin axons make contacts with SCs as early as 3 dpi but that YAP/TAZ are not upregulated, whereas there is obvious upregulation at 6 dpi when many of these axons become thicker (i.e., >1um or large enough to be myelinated). These observations are consistent with our earlier finding that YAP/TAZ are expressed only in myelinating, but not in non-myelinating, SCs (Grove et al., 2017), and further support the notion that the absence and presence of axons determine YAP/TAZ down- and upregulation in SCs after injury, respectively. We believe that this additional data further justify our suggestion in the discussion that the same mechanism may maintain YAP/TAZ expression in adult SCs in intact nerve.

It is also noteworthy that, despite drastic phenotype differences, both innervated and denervated SCs and their processes reside within basal lamina (BL) tubes. It is also well established that axons degenerate and regenerate within BL tubes, losing or regaining contacts with SCs and SC processes that tightly fill the BL tubes and comprise the bands of Büngner. Therefore, although the confocal microscopy that we used in the present study does not provide sufficient resolution to demonstrate direct cell-cell contact between SCs and the fluorescently labeled axons, the presence of axons closely adjacent to the SC nuclei of interest strongly suggests that these SCs have made direct contact with axons.

7) In Discussion paragraph four, the authors propose that "YAP/TAZ powerfully stimulate developing and adult SCs to proliferate at all stages". This conclusion is not directly supported by data. The authors point out that in a recent study (Mindos et al., 2017) YAP and TAZ are critically regulated by Merlin in Schwann cells (SCs), and inactivation of Merlin leads to higher levels of YAP and TAZ preventing axon regeneration and remyelination. However, it remains unclear how YAP (in Taz iKO nerves) can myelinate regenerated axons. Would removal of Merlin in Taz iKO nerves also prevent myelination? An experiment that identifies these mechanisms would help.

We found that SCs do not require YAP/TAZ to proliferate after nerve injury. This finding was quite unexpected because YAP/TAZ are crucial for the proliferation of developing SCs (Grove et al., 2017). In our discussion of this interesting finding, we suggested that the finding was unlikely to indicate that YAP/TAZ are unable to stimulate proliferation of “mature” SCs, which was what we meant by our statement that *"*YAP/TAZ…stimulate…adult SCs to proliferate…". We referred to two recent publications from other groups which support the statement, and which reported increased proliferation of mature SCs in response to abnormally high activation of YAP in injured nerve in the absence of Merlin (Mindos et al., 2017) or of YAP/TAZ in intact nerve in the absence of LATS1/2 (Wu et al., 2018). We think that these reports provide sufficient rationale for our statement and discussion.

It is worth noting that both reports found increased YAP-, or YAP/TAZ-dependent SC proliferation when the upstream regulators of YAP/TAZ (Merlin, LATS1/2) were experimentally dysregulated, whereas we observed YAP/TAZ-independent proliferation of SCs when these signals are intact and properly regulating spatiotemporal expression of YAP/TAZ after injury. As we mentioned in #5 response, we postulate that active regulatory mechanism(s) remove YAT/TAZ from SCs to help dedifferentiation upon axon degeneration after injury, and then upregulate them upon axon regeneration to induce remyelination. We also postulate that the regulatory mechanism(s) prevent redifferentiating SCs that contact large regenerating axons from excessively upregulating YAP/TAZ, because excessive activation of YAP/TAZ may stimulate SC proliferation rather than differentiation, as in Merlin- or LATS1/2-null mutants.

Consistent with this notion, we also found that YAP expression remains at a normal level in Taz iKO and that YAP alone promote myelination in the absence of TAZ, although YAP and TAZ together induce more efficient remyelination. We are quite confident to conclude that YAP and TAZ are functionally redundant in remyelination. We do not know why the reviewer questions our conclusion, but speculate that it is related to the report of Mindos et al., 2017. These investigators examined nerve injury in the absence of Merlin and raised the possibility that YAP and TAZ may have different functions, which contradicts our conclusion. This report also included some other discrepancies with our observations, such as TAZ expression in denervated SCs. We are very interested to know if Merlin indeed regulates YAP and TAZ expression differently, the mechanisms by which Merlin regulates YAP/TAZ, and how its loss impacts spatiotemporal expression and thus the activity of YAP/TAZ. This information will be of great interest, but, in our opinion, generation and in-depth analysis of Merlin and Taz double mutants are beyond the scope of the present work.

8) The authors should check the statistics they applied. A t-test assumes that samples are normally distributed. This is not likely the case for 3 replicates. Use the Mann-Whitney test as alternative. Western blots should also be quantified and for all quantifications (imaging, Western blotting) individual data points should be shown.

As suggested, we used the Mann-Whitney U test, quantified additional Western blots (Figure 4—figure supplement 1, Figure 8A) and indicate individual data points on all graphs.

9) Figure 4: Does the loss of YAP/TAZ result in a delay of axonal re-generation? Mindos et al., 2017, had shown in mice hat regenerating axons could reach up to 13-14 mm distal to the injury site within 7 days of nerve crush. The present study accessed regeneration 12-14 days after injury but only within the 5-10 mm distal region, a paradigm unlikely to detect a delay in axonal regeneration if present. To address this issue, regeneration should be analyzed earlier and ideally along its time course.

As requested, we analyzed an earlier phase of regeneration by selectively labeling regenerating axons at 3 days after injury, and the result is presented in the new Figure 6. We observed similar numbers of axons that penetrated the injured site and reached ~4mm distal to the crushed site in YAP/TAZ iDKO, as in WT mice. Thus, it is highly unlikely that regeneration is delayed in the absence of YAP/TAZ. We chose 3 days after injury to assess the onset and early phase of regeneration when axotomized distal axons are actively degenerating. In rodents axons regenerate along the distal stump at a speed of up to 4mm/day, and even the most distal muscles are reinnervated in less than 12 days in mice. We believe, therefore, that if we were to find a difference of 1-2 millimeters in how far the longest axons regenerate at times later than 3 dpi in YAP/TAZ iDKO and WT mice, that this difference would not reliably indicate poor regeneration.

10) Similarly Figures 6, 7: Nerve regeneration/remyelination was accessed 12-13 dpi, a couple of days before death of the mice is expected. As stated in subsection “SCs lacking YAP/TAZ convert to repair SCs and support axon regeneration”, iDKO mice die about 14 days after tamoxifen injection (also stated in Grove et al., 2017). At the time point analyzed, even uninjured iDKO mice exhibit clear PNS nerve defects (shivering, decrease in nerve conduction velocity, demyelination etc) as well as ataxia and impaired breathing (Grove et al., 2017). This complicates interpretation of the data on remyelination as the failure to re-build myelin could be due to a secondary effect stemming from the poor physiological condition of the mice. To prove that the remyelination failure is indeed due to the loss of YAP/TAZ, one would like to see a rescue experiment that restores the YAP/TAZ function in the iDKO Schwann cells. Unless such experiments are provided, the conclusion and discussion must be carefully rephrased.

We now discuss why we consider it highly unlikely that the remyelination failure is due to poor physiological condition of the iDKO mice. First, axons regenerate normally in iDKO, which is unlikely if SCs are selectively vulnerable to poor physiological condition. Furthermore, iDKO SCs proliferate and trans-differentiate to repair-SCs normally. iDKO SCs also downregulate c-Jun to prepare for remyelination, whereas they maintain a higher level of Oct 6 than WT SCs, consistent with the failure of iDKO SCs to upregulate Krox 20 and MBP. Second, iDKO SCs wrap around individual axons, but fail to myelinate them, indicating that they proceed to the promyelination stage but no further. Therefore, one would have to postulate that the poor physiological condition of iDKO mice has a very specific effect on a particular remyelination stage, which we find unlikely.

[Editors' note: further revisions were suggested prior to acceptance, as described below.]

Reviewer #1:The revised version of the manuscript has significantly improved as compared to the original study. Nevertheless, the authors have not performed the requested additional experiments, which are instead argued in the rebuttal letter.While it is clear from the presented data that p-YAP relocalizes in cytoplasm (Figure 2D), in Figure 1B the authors show and state that " the dramatic down- and up- regulation of YAP/TAZ concurrent with axon degeneration and regeneration, respectively, suggest that SCs are dependent on axons from YAP/TAZ nuclear localization". Given the experimental data it would be more correct talking about expression rather than localization. The latter indeed assumes a shuttle between in an out of the nucleus, which is valid for p-YAP and also in that case only partially.

We agree with the reviewer and modified the statement by changing “localization” to “expression”.

Reviewer #2:Grove et al. provide a detailed point-to-point response that satisfactorily discusses all issues raised.However, this reviewer still has the following concerns related to the quantification of WB data (see below). Once clarified, the manuscript can be granted for publication.Figure 1C, Figure 1—figure supplement 2:For the quantification, 4 data points are shown per condition. supplemental data show 2 uncropped blots that are 2 different exposures from the very same gel as presented in 1C (as the authors also specify). However, 3 independently performed blots that should show independently processed samples and were used for the quantification are not available. The authors need to provide the correct source data. Also note that Figure 1—figure supplement 2 contains a staining of nerve fibers, but no WB data.

There seems to have been confusion in supplement figure numbers. It was Figure 1—figure supplement 1 where we previously presented WB data (i.e., 2 exposures of the same gel in 1C). Figure 1—figure supplement 2 was meant to contain no WB data. The figure that the reviewer requests that we amend is Figure 1—figure supplement 1, not Figure 1—figure supplement 2.

As requested, Figure1—figure supplement 1 now shows additional, uncropped Western Blots from all the mice used for quantification at 4 different time points. They are from independently processed samples from each mouse: 3 mice (1 dpi, 6 dpi), 6 mice (3 dpi) and 4 mice (12 dpi).

Figure 4—figure supplement 1 and 2:The quantification implicates two blots that were quantified. However, supplement 2 contains the same blot as presented in supplement 1. The authors need to provide the correct source data.

As requested, Figure 4—figure supplement 2 now show uncropped blots from two WT and two iDKO mice, independently processed for quantification.

Figure 8A, Figure 8—figure supplement 2:The quantification implicates 3 blots that were quantified. However, supplement 2 contains the same blot as presented in Figure 8A. Again, the authors need to provide the correct source data for all blots quantified.

As requested, Figure 8—figure supplement 2 now shows uncropped blots of 3 WT and 3 Taz iKO mice, independently processed for quantification.

Reviewer #3:This is a revised manuscript which investigates the role of YAP/TAZ in mediating Schwann cell response to injury and PNS regeneration. The authors have addressed reviewers concerns sufficiently, by either conducting additional experiments or in writing. Overall, data from the study is solid and the findings are potentially significant to the field of PNS injury and regeneration. I only have one concern, which can be addressed in writing by the authors.Figure 1 and Figure 1—figure supplement 1 (paragraph four subsection “YAP/TAZ expression in Schwann cells is axon-dependent”). I am still confused about the discrepancy between the WB and IHC results on YAP expression in denervated nerves. When detected WB, YAP expression continues to decrease after nerve crush and does not recover, whereas when assessed by IHC, YAP expression returns at 6 and 12D. The authors claim that the WB result reflects YAP expression in other cell types (I have to say, I do not agree with the explanation). However, they also make a point in the Discussion that WB was done using lysates prepared from nerves that had been de-sheathed (epi- and perineurium removed) to ensure enrichment of SC-derived proteins.Since most of the YAP is nuclear at 6 and 12D, shown by the IHC, is it possible that the lysis buffer used for WB does not recover nuclear YAP in the Schwann cells, thus the WB only reflects the cytoplasmic YAP?

For all WBs, sciatic nerves were lysed in RIPA buffer, which is a stringent buffer. Nevertheless, for all antibodies used in WBs, we performed pilot experiments where the sciatic nerve pellet left over after RIPA extraction was dissolved in 3xSDS-PAGE loading buffer (this buffer contains 6% SDS). An equal fraction of the pellet and lysate was run on SDS-PAGE to ensure that all of the protein under study was in the lysate fraction, which was the case with YAP and TAZ. Hence, we are confident that all of the nuclear and cytoplasmic YAP in Schwann cells was in the lysate used for WBs.

IHC revealed numerous SCs exhibiting YAP apparently in their nuclei at 6D or later. However, quantitative analysis was difficult to perform with the sole antibody that specifically recognized YAP (but not TAZ) in IHC. Accordingly, we claim that YAP is upregulated in SCs, but we do not intend to imply that its levels return to normal in SCs. It was indeed our impression that YAP staining intensity may remain weak possibly in quite many SCs even at 24D. We will be interested to explore this possibility, when additional antibodies of high quality are available for YAP or TAZ specific IHC.